**Letter**

# Cancer-independent somatic mutation of the wild-type *NF1* allele in normal tissues in neurofibromatosis type 1

Thomas R. W. Oliver [1,2,11], Andrew R. J. Lawson [1,11], Henry Lee-Six [1,2,11], Anna Tollit[3], Hyunchul Jung[1], Yvette Hooks[1], Rashesh Sanghvi [1], Matthew D. Young [1], Timothy M. Butler [1], Pantelis A. Nicola[1], Taryn D. Treger[1,2,4], Stefanie V. Lensing[1], G. A. Amos Burke [2,5], Kristian Aquilina[6], Ulrike Löbel[6], Isidro Cortes-Ciriano [7], Darren Hargrave[6,8], Mette Jorgensen[6], Flora A. Jessop[2], Tim H. H. Coorens [9], Adrienne M. Flanagan [3,10], Kieren Allinson [2,12] ✉, Inigo Martincorena [1,12] ✉, Thomas S. Jacques [6,8,12] ✉ & Sam Behjati [1,2,4,12] ✉

Cancer predisposition syndromes mediated by recessive cancer genes generate tumors via somatic variants (second hits) in the unaffected allele. Second hits may or may not be sufficient for neoplastic transformation. Here we performed whole-genome and whole-exome sequencing on 479 tissue biopsies from a child with neurofibromatosis type 1, a multisystem cancer-predisposing syndrome mediated by constitutive monoallelic *NF1* inactivation. We identified multiple independent *NF1* driver variants in histologically normal tissues, but not in 610 biopsies from two nonpredisposed children. We corroborated this finding using targeted duplex sequencing, including a further nine adults with the same syndrome. Overall, truncating *NF1* mutations were under positive selection in normal tissues from individuals with neurofibromatosis type 1. We demonstrate that normal tissues in neurofibromatosis type 1 commonly harbor second hits in *NF1*, the extent and pattern of which may underpin the syndrome's cancer phenotype.

In recessive tumor predisposition syndromes, one allele is mutated in the zygote (or, rarely, in early embryogenesis), while the second allele is inactivated by subsequent somatic mutation (second hit; Fig. 1a). Although a second hit would ordinarily be expected to lead to neoplasia, it is possible that some cells remain phenotypically normal in the presence of biallelic mutation, just as oncogenic mutations have been reported within healthy tissues. Recent studies of normal adult tissues have revealed bona fide cancer-causing (driver) mutations that accumulate with age and exposure to environmental mutagens, primarily in exposed epithelial tissues[1–4]. The acquisition of mutations in normal tissues may be accelerated by germline mutations perturbing the fidelity of DNA replication, as seen in normal intestinal crypts of patients with a mutant DNA polymerase[5]. Furthermore, in children with malignant rhabdoid tumors (cancers driven by biallelic inactivation of *SMARCB1*), we have observed normal tissues that share a genetic ancestor with the nearby tumor and harbor the same somatic *SMARCB1* hit, without an elevated mutation rate[6] (Fig. 1b). We therefore speculated that second hits may occur in normal tissues of predisposed individuals that are unrelated to tumor lineages or affected by hypermutation, which we set out to investigate here (Fig. 1c).

Neurofibromatosis type 1 is a complex multisystem disorder that predisposes to neoplasia. It is caused by germline mutation in the *NF1*

gene, a tumor suppressor gene that encodes neurofibromin, a negative regulator of intracellular RAS/MAPK signaling. The syndrome's neoplastic phenotype is variable and tends to affect neuroectodermal lineages, although tissues derived from other germ layers also have an increased risk of cancer. An essential diagnostic feature of neurofibromatosis type 1 is the café au lait spot, a macroscopically visible clonal expansion of melanocytes[7,8]. Other neoplastic manifestations, which exhibit variable penetrance, include neurofibromas, skeletal dysplasias, leukemias, malignant peripheral nerve sheath tumors and gliomas[9–13]. In all these lesions, *NF1*, as a recessive cancer gene, exhibits a second mutation, not infrequently as the sole detected somatic driver event[11,12], consistent with Knudson's two-hit hypothesis[14].

We performed a postmortem study of three children aged <10 years old with high-grade midline gliomas—two (PD50297 and PD51123) with sporadic tumors (*H3F3A* K27M mutant) and one with neurofibromatosis type 1 with a pathogenic truncating *NF1* (c.3113 + 1G>A) germline mutation. Our key question was whether normal tissues across the body harbored driver events, in particular in the predisposed child. We extensively sampled normal tissues and neoplasms (Supplementary Tables 1 and 2), which, in the case of the child with neurofibromatosis, included a brain tumor, a subcutaneous spindle cell lesion (Extended Data Fig. 1) and a café au lait spot. Guided by parental wishes, we surveyed central nervous system (CNS) tissues in all three children and extracranial tissues in two of them, including the predisposed child. None of the children had been pretreated with cytotoxic chemotherapy. Radiotherapy was given to the two children with sporadic tumors.

In total, we performed whole-genome sequencing (WGS) on 838 microdissected groups of cells (median coverage 28×), using an established approach that we and others have pursued in the study of normal tissue genomes[15] (Fig. 1d,e). We supplemented this with additional bulk tissue WGS (*n* = 71) and whole-exome sequencing (WES) of other microdissected tissues (*n* = 180; Fig. 1e). We assembled catalogs of all classes of mutations (substitutions[16], insertions and deletions (InDels), rearrangements and copy number changes; Supplementary Tables 3–8 and Supplementary Note) using a validated variant calling pipeline (Methods; Supplementary Note). To exclude low-level tumor contamination of normal tissues, we quantified the extent of tumor infiltration in each sample (including samples distant from the tumor) by searching for the mutations assigned to the tumor's phylogenetic trunk (Supplementary Note and Supplementary Tables 3 and 9).

We identified somatic driver variants in both neoplastic and normal tissues (Fig. 2 and Supplementary Tables 10–14). Gliomas exhibited a multitude of driver mutations in cancer genes known to operate in gliomagenesis[17,18]. The normal tissues of the two nonpredisposed children bore comparatively few cancer-associated mutations, yielding only a *CREBBP* frameshift mutation (p.Q2199fs*99) within a single colonic crypt (PD50297g_lo0012) by our standard pipeline. Further inspection of the copy number data revealed one more putative driver, chromosome 11p loss of heterozygosity (LOH; Extended Data Fig. 2) in a nerve (PD51123t_lo0028), a variant commonly reported in Wilms tumor, rhabdomyosarcoma and hepatoblastoma[19].

By contrast, in the child with neurofibromatosis type 1, we found bona fide somatic *NF1* driver point mutations (Fig. 2) that either truncated the gene (p.R304*, p.K233fs*48 and p.I679fs*21) or were likely oncogenic based on recurrence, in silico predictions, correlation between genotype and phenotype and functional studies (p.Y489C)[20,21]. These mutations were detected by a combination of WGS and WES of microdissections or bulk tissues. They occurred in anatomically distant regions of the CNS (left parietal cortex, cerebellar hemisphere or spinal cord). The affected tissues appeared macroscopically and microscopically normal and were not correlated with focal areas of signal intensity on imaging (a feature often found in the brains of children with neurofibromatosis type 1 (ref. 22; Extended Data Figs. 3 and 4). The variant allele frequencies (VAFs) of second *NF1* hits indicated clone sizes as

large as 56% of cells (clone size = 2 × VAF) in a microdissection (hundreds of cells) and 19% in a bulk tissue (a macroscopic piece of tissue). *NF1* second hits were independent of those found in the clonal lesions (glioma, spindle cell lesion and café au lait spot; Fig. 3a). In particular, as both copies of *NF1* in the glioma were already inactivated, there should be no selection pressure for further loss-of-function mutations in *NF1* in tumor cells. Given this and our ability to detect tumor contamination accurately (Supplementary Note), the *NF1* mutations in normal tissues are not the result of tumor cells infiltrating normal tissues. The mutation burden of *NF1* null normal tissues was inconsistent with a recent clonal expansion (Fig. 3b; Methods). Like the spindle cell lesion and café au lait spot, no additional driver mutations were identified in the *NF1* null histologically normal tissues.

To expand the breadth and depth of *NF1* mutations detected within the normal tissues of the predisposed child, we used two strategies. First, we re-examined all the child's sequencing data for evidence of LOH of the *NF1* locus (chromosome 17q). As this child's glioma harbored complete LOH of chromosome 17 (Fig. 3a), we were able to increase the sensitivity to detect allelic shift by obtaining definitive chromosome 17 haplotypes from phasing allele-specific single nucleotide polymorphisms (Methods; Supplementary Tables 3 and 4 and Extended Data Fig. 5). This haplotype-resolved copy number calling revealed six instances of LOH in normal tissues (Fig. 2), of which at least two were demarcated by distinct breakpoints consistent with independent events (Extended Data Fig. 5). We captured LOH-driven *NF1* null clone sizes as small as 2% (PD51122b; cerebellum) and up to 13% (PD51122h_lo0012; occipital cortex), and no second *NF1* hit was related to a neoplasm. Curiously, while this approach did not yield any *NF1* null clones in any non-neuroectodermal lineage, it did identify rare examples where the germline mutant allele was lost in microdissections of tissues derived from other germ layers (bladder muscle (PD51122s_lo0012) and a renal tubule (PD51122u_lo0009); Extended Data Fig. 6). All *NF1* second hits identified thus far are shown in Fig. 3c.

Second, to deepen our search for *NF1* null normal cells, we assessed all samples for evidence of *NF1* point mutations that had been previously called in normal tissues. Three variants (p.R304*, p.I679fs*21 and p.Y489C) were detectable in more than one biopsy above the locus-specific error rate (Methods; Fig. 3d and Supplementary Table 15). There were the following two possible explanations: either the shared variants arose before the seeding of the anatomical areas in which the mutations were found, or mutations appeared independently in different tissues. We could establish which scenario was more likely by comparing the total number of mutations across the genome that were shared between affected tissues—those with a common developmental root would possess more. For *NF1* mutations that spanned only regions of the brain (p.R304* and p.I679fs*21), affected tissues shared significantly more mutations with each other than unaffected brain regions did (*P* < 0.001 for both mutations, one-sided permutation test; Fig. 3e; Methods), implicating a common ancestor in their development, although convergent evolution with shared selection pressures between developmentally related tissues is also possible. The same could not be said for the *NF1* mutation found in both the brain and spleen when compared to normal tissues of the CNS and mesoderm (p.Y489C; *P* = 0.373, one-sided permutation test; Fig. 3d), meaning that they likely developed independently.

Taken together, the multiple lines of inquiry we had pursued thus far pointed toward the enrichment of *NF1* nonsynonymous mutations within the normal tissues of a predisposed individual but not wild-type individuals, at least within the brain and spinal cord. It seems likely that this pattern of *NF1* mutation has emerged as a consequence of positive selective pressure, given the absence of the concomitant silent and intronic *NF1* variants (Fig. 2) that would be expected under a neutral model.

To establish statistically whether there is a positive selection for nonsynonymous variants in *NF1* in predisposed normal tissues, we

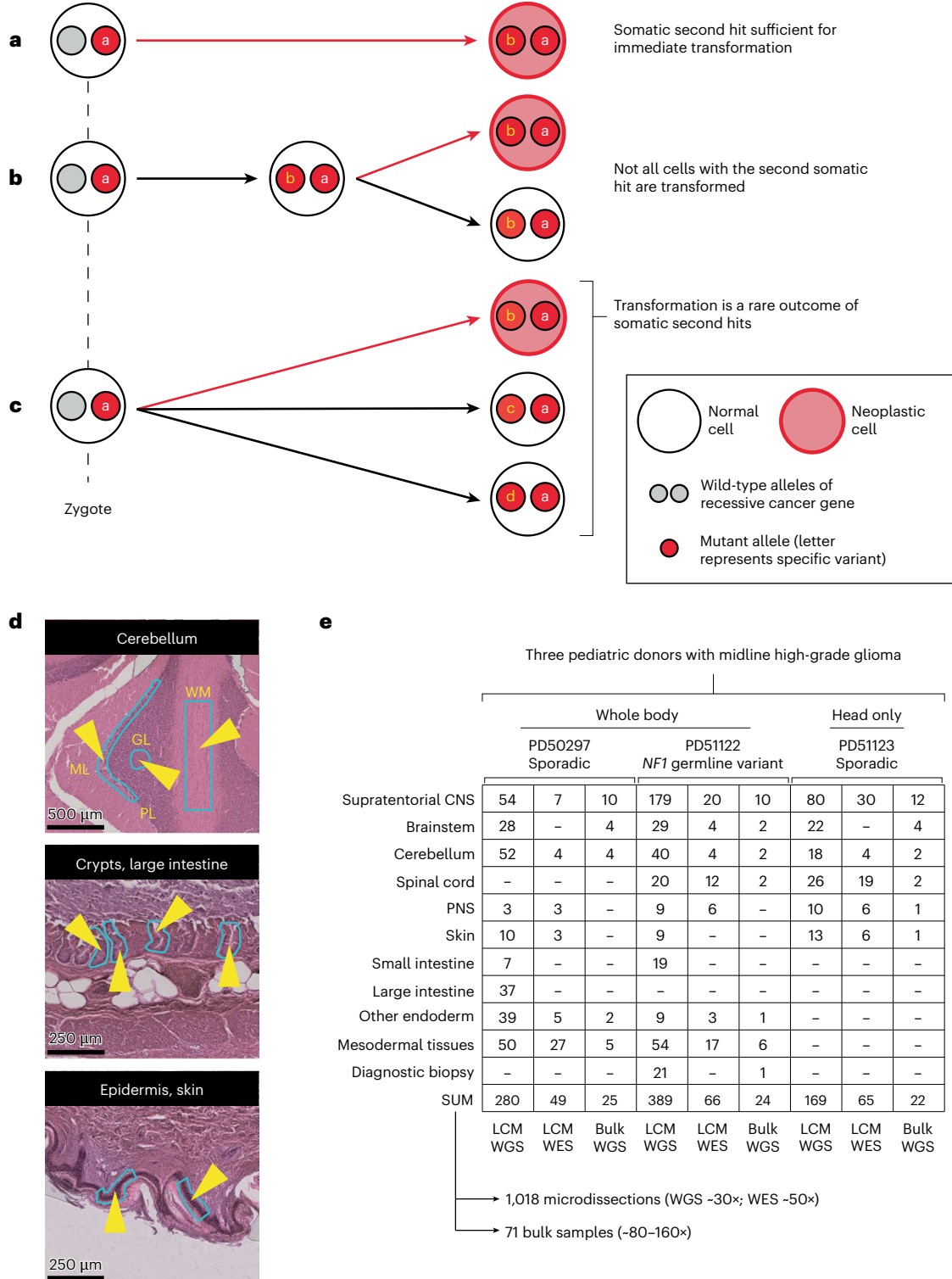

**Fig. 1 | Concepts and experimental approach. a**, The second mutation in a recessive tumor predisposition syndrome is typically thought to lead to neoplasia. **b**, Some second hits may be found in the adjacent normal tissue to a childhood cancer, indicating that their presence is insufficient for neoplastic transformation. **c**, The possibility remains that second hits may be sustained in normal tissues that are independent of the cancer cell lineage. **d**, Histological images of three illustrative microdissected tissues are shown. The layers of the cerebellar cortex are annotated on the uppermost image. The light blue outlines with yellow arrowheads on the images are representative regions microdissected. Scale bars = 500, 250 and 250 μm (top to bottom). **e**, Experiment overview, detailing the number of bulk- and LCM-derived sequences generated per anatomical region per child. Please note that this includes all biopsies, irrespective of tumor involvement. A version of this table, limited to the tumor biopsies used in the high-grade glioma driver mutation identification, is provided as Supplementary Table 2. ML, molecular layer; PL, Purkinje layer; GL, granular layer; WM, white matter; CNS, central nervous system; PNS, peripheral nervous system; LCM, laser capture microdissection.

| | | | Child with *NF1* germline mutation (PD51122) | | | | | | Children without *NF1* germline mutation (PD50297 and PD51123) | | | | | |
| | | | | *NF1* variants | | | | | | *NF1* variants | | | | |
| | | | Number of samples | Nonsynonymous | Silent | Intronic | Structural* | Genes with other driver events** | Number of samples | Nonsynonymous | Silent | Intronic | Structural* | Genes with other driver events** |
|---|---|---|---|---|---|---|---|---|---|---|---|---|---|---|
| Neoplasms | High-grade glioma | Bulk WGS | 7 | | | | | CDK6, CDKN2A, EGFR, KIT, KRAS, MYCN, PDGFRA, PIK3CA, PIK3R1 and TP53 | 11 | | | | | PD50297: ATRX, CCND2, H3F3A, PTEN, RB1, TP53 PD51123: H3F3A, KDR, KIT, PDGFRA, PTEN and TP53 |
| | | LCM WGS | 63 | 0 | 0 | 0 | LOH | | 73 | PD50297: p.T569fs*14 | 0 | 7 | PD50297: Del PD51123: Inv | |
| | | LCM WES | 0 | | | | | | 1 | | | | | |
| | Spindle cell lesion | Bulk WGS | 0 | | | | | None | – | | | | | – |
| | | LCM WGS | 1 | 0 | 0 | 0 | Del | | – | – | – | – | – | |
| | | LCM WES | 0 | | | | | | – | | | | | |
| | Café au lait macule | Bulk WGS | 0 | | | | | None | – | | | | | – |
| | | LCM WGS | 9 | 0 | 0 | 0 | LOH | | – | – | – | – | – | |
| | | LCM WES | 0 | | | | | | – | | | | | |
| Normal neuroectodermal tissues | Supra-tentorial CNS | Bulk WGS | 2 | | | | | None | 13 | | | | | None |
| | | LCM WGS | 106 | p.R304* | 0 | 0 | 3 × LOH* | | 94 | 0 | 0 | 0 | 0 | |
| | | LCM WES | 17 | | | | | | 32 | | | | | |
| | Brainstem | Bulk WGS | 2 | | | | | None | – | | | | | – |
| | | LCM WGS | 7 | p.I679fs*21 | 0 | 0 | 0 | | – | – | – | – | – | |
| | | LCM WES | 3 | | | | | | – | | | | | |
| | Cerebellum | Bulk WGS | 2 | | | | | None | 2 | | | | | None |
| | | LCM WGS | 38 | p.Y489C | 0 | 0 | 2 × LOH* | | 42 | 0 | 0 | 0 | 0 | |
| | | LCM WES | 4 | | | | | | 4 | | | | | |
| | Spinal cord | Bulk WGS | 1 | | | | | MSH6*** | 0 | | | | | None |
| | | LCM WGS | 20 | p.K233fs*48 | 0 | 0 | LOH* | | 24 | 0 | 0 | 0 | 0 | |
| | | LCM WES | 12 | | | | | | 18 | | | | | |
| | Nerves/nerve roots | Bulk WGS | 0 | | | | | None | 1 | | | | | None**** |
| | | LCM WGS | 20 | 0 | 0 | 0 | 0 | | 18 | 0 | 0 | 0 | 0 | |
| | | LCM WES | 8 | | | | | | 13 | | | | | |
| Normal non-neuroectodermal tissues | Skin (ectoderm) | Bulk WGS | – | | | | | – | 1 | | | | | None |
| | | LCM WGS | – | – | – | – | – | | 20 | 0 | 0 | 0 | 0 | |
| | | LCM WES | – | | | | | | 9 | | | | | |
| | Spleen (mesoderm) | Bulk WGS | 1 | | | | | MSH6*** | 1 | | | | | None |
| | | LCM WGS | 13 | 0 | 0 | 0 | 0 | | 8 | 0 | 0 | 0 | 0 | |
| | | LCM WES | 6 | | | | | | 6 | | | | | |
| | Other mesoderm | Bulk WGS | 5 | | | | | None | 4 | | | | | None |
| | | LCM WGS | 37 | 0 | 0 | 0 | 0 | | 51 | 0 | 0 | 0 | 0 | |
| | | LCM WES | 11 | | | | | | 20 | | | | | |
| | Endoderm | Bulk WGS | 1 | | | | | None | 2 | | | | | CREBBP |
| | | LCM WGS | 22 | 0 | 0 | 0 | 0 | | 66 | 0 | 0 | 0 | 0 | |
| | | LCM WES | 3 | | | | | | 2 | | | | | |

**Fig. 2 | *NF1* mutations and driver events identified by sequencing bulk tissues or microdissections.** Individual driver variants can be found in the lists of mutations provided in Supplementary Tables 10–14. A single asterisk indicates that the identification of the LOH event was only possible because we could phase parental alleles (Methods). The low cell fraction of many of these meant that it was not possible to determine the breakpoint. When counting LOH events, those called from sequences derived from the same original bulk biopsy are treated as the same event, and those from different biopsies are treated as unique. These events should not be considered when comparing mutations against the other two children because their SNP alleles on chromosome 17 could not be phased. Double asterisks indicate that in sequences from the high-grade glioma, we only considered mutations in genes recognized in a large meta-analysis to be drivers of these neoplasms (Methods). Triple asterisks indicate that one additional *MSH6* frameshift mutation (p.F1088fs*2) of the child with neurofibromatosis type 1; this, however, remained heterozygous without evidence of hypermutation and did not co-occur with somatic *NF1* mutation, making it of uncertain significance. Four asterisks indicate that chromosome 11p LOH was identified in a single sample after manual inspection of the copy number output, although it was too low a fraction to be detected by the copy number caller. Del, deletion; Inv, inversion.

re-interrogated 60 normal tissue samples (21 from the predisposed child and 39 from the unaffected children) by duplex sequencing of the *NF1* gene[23,24]. Duplex sequencing, through barcode tagging of both strands of DNA molecules, enables highly sensitive and specific mutation calling, which may deliver sufficient variants for formal statistical assessment of selection through the nonsynonymous to synonymous variant ratio ($dN/dS$)[17]. Duplex sequencing yielded a total of 29 nonsynonymous (21 truncating; eight missense or in-frame) and two synonymous *NF1* mutations in the predisposed child (Supplementary Table 16). By contrast, in the normal tissues of the two children without neurofibromatosis, we detected nine nonsynonymous (four truncating; five missense or in-frame) and five synonymous *NF1* mutations (Supplementary Table 16). Calculation of the $dN/dS$ ratio provided strong evidence of positive selection for truncating *NF1* variants in normal tissues from the child with a germline predisposition (Fig. 3f).

Interestingly, the spleen had a particularly high proportion of truncating variants, and, when analyzed separately from other tissues, it had the highest $dN/dS$ ratio (Fig. 3f). This finding was of interest given that neurofibromatosis type 1 predisposes to juvenile myelomonocytic leukemia, which always (as per the diagnostic definition) involves the spleen[25], whether through entrapment of leukemic cells or as their organ of origin.

Next, we extended our analysis into normal tissue from adults with neurofibromatosis type 1 (Supplementary Table 17). We were able to obtain normal peripheral nerves, muscle tissue or blood from nine individuals who underwent extensive surgical resections for sarcoma. The principal cellular material of peripheral nerves is made up of Schwann cells that are derived from the neuroectoderm, whereas muscle and blood develop from mesoderm. Consistent with the pattern of mutation we observed in pediatric tissues, we found, by duplex

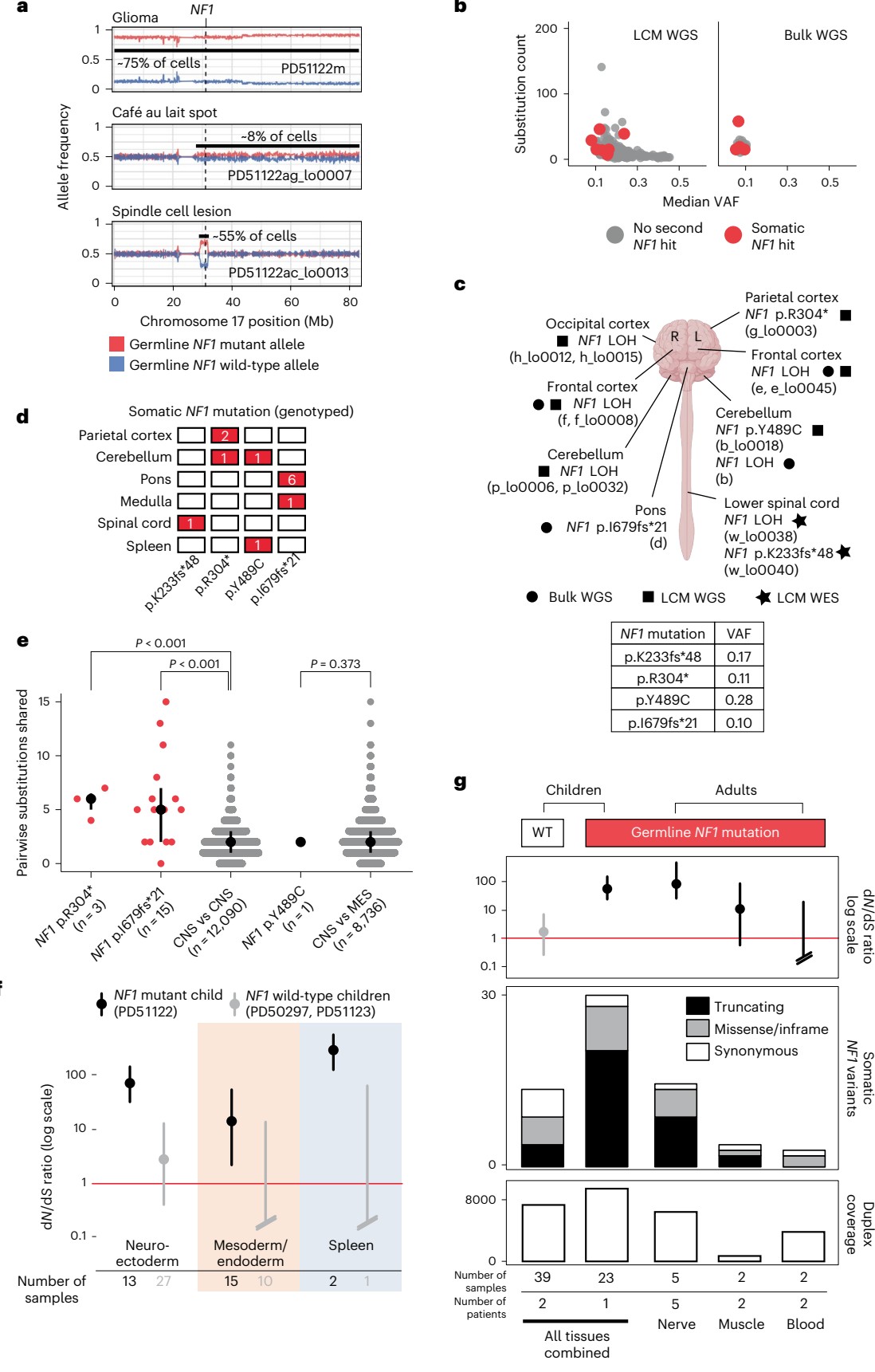

sequencing, a stark excess of truncating *NF1* mutations in peripheral nerves, indicative of positive selection (Fig. 3g).

In this study of *NF1* mutations in individuals with neurofibromatosis type 1, we observed independent second *NF1* hits in macroscopically

and histologically normal pediatric and adult tissues. Multiple lines of evidence arrive at the same conclusion: in neurofibromatosis type 1, nonsynonymous second somatic mutations of *NF1* are selected for in histologically normal tissues. Although *NF1* is a ubiquitously

**Fig. 3 | *NF1* null clones pervade the normal tissues of individuals with neurofibromatosis type 1. a**, Loss of the wild-type *NF1* allele is an independent event in each of the three neoplasms in PD51122. The allele frequency (*y* axis) represents a rolling window of 50 SNPs. Gridlines are present for matching to coordinates. **b**, The substitution count and median VAF in normal brain, color-coded by the presence of biallelic *NF1* mutation. **c**, Schematic representation of the brain and spinal cord outlining the location of each somatic *NF1* mutation in normal tissue from PD51122 discovered by different sequencing methods (before genotyping). The VAF of each mutation is shown in the table (2sf). The figure is created with BioRender.com. **d**, Distribution of somatic *NF1* mutations across tissues from PD51122 (above the locus-specific error rate; Methods). The number within the box indicates the number of mutated samples from that tissue (red if >0). **e**, Pairwise comparison of the number of substitutions shared between whole-genome sequences of normal biopsies, according to whether they possess the same *NF1* second hit. The black line represents the interquartile range. The

black dot is the median. *P* values were generated using one-sided permutation tests (Methods). '*n*' refers to the number of pairwise comparisons, not samples. The CNS and MES groups exclude normal tissues with a second *NF1* hit. **f**, The d*N*/d*S* ratios for truncating variants, according to germ layer and *NF1* germline mutation status. The dot represents the maximum likelihood estimate, and the lines represent the 95% credible interval. When the lower bound of the credible interval is above 1 (red line), there is a statistically significant positive selection. Credible intervals falling below the boundary of the plot are terminated with slanted double lines. **g**, Normal tissues from adults with neurofibromatosis type 1 are grouped by tissue type and evaluated for an excess of nonsynonymous variants in *NF1* and compared with the index children. Top, d*N*/d*S* ratios for truncating mutations (the dot represents the maximum likelihood estimate, and the lines represent the 95% credible interval); middle, counts of variants in *NF1*; bottom, total duplex coverage (Methods; Supplementary Note) over *NF1* in each group. R, right; L, left; MES, mesoderm; WT, wild type.

expressed gene, the tissue distribution of neoplasms associated with neurofibromatosis type 1 is not random, showing a predilection for neuroectodermal lineages[26], which is mirrored in the distribution of second *NF1* hits we identified. Unlike our study, though, where these mutations pervaded the CNS, most brain tumors that arise in children with neurofibromatosis type 1 are localized to the optic pathway and brainstem[26,27]. Our findings may thus explain some, but not all, of the cancer phenotypes associated with neurofibromatosis type 1.

Three factors, unrelated to the germline mutation status of *NF1*, may also have contributed to the number of nonsynonymous mutations we observed. The first is age, as neutral mutations accrue with time, and the second is the size of the *NF1* gene. *NF1* has one of the largest footprints of any gene (8,520 bp coding sequence compared to a median length across human genes of 1,257 bp (ref. [28])), meaning there are simply more sites to mutate. Hypermutation is a third factor that might augment the rate of driver mutation acquisition, but our comprehensive study of the index case found no evidence to support that here. Notably, all these factors are accounted for by our model of selection (d*N*/d*S*), meaning that they cannot explain our data. We can assume, then, that the strength of the signal we see in the carriers compared to unaffected children is the result of selective pressure specifically for second hits. The fact that this is so readily apparent in the extensively studied child suggests that this occurs from an early age, possibly even during development (Fig. 3be). Given the extent to which we observed *NF1* loss-of-function variants in normal tissues, it seems reasonable to propose that although in certain contexts second hits may be sufficient to cause neoplasia[10,11], as suggested by our case's café au lait spot and spindle cell lesion, transformation to a discernible tumor is an uncommon immediate outcome of biallelic *NF1* loss.

This finding may represent a fundamental principle of *NF1* mutation in neurofibromatosis type 1, of which determining the precise nuances and clinical implications will require extensive surveys in human tissues[29]. Consistent with our data, mouse models support a complex relationship between cellular genotype and phenotype in neurofibromatosis type 1—the genetic background of the mouse, the identity of the cell in which *NF1* is inactivated, the presence of cooperating somatic mutations and the status of *NF1* function in neighboring cells, all appear to affect tumor development[30,31]. From a practical, clinical point of view, it is conceivable that the extent of second *NF1* hits in normal tissues represents a quantifiable link between germline genotype and cancer risk. In the broader context of recessive cancer predispositions, our findings call for systematic investigations to establish whether second hits occur commonly in such predispositions or delineate a particular group of syndromes.

## Online content

Any methods, additional references, Nature Portfolio reporting summaries, source data, extended data, supplementary information, acknowledgements, peer review information; details of author contributions

and competing interests; and statements of data and code availability are available at https://doi.org/10.1038/s41588-025-02097-2.

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

¹Wellcome Sanger Institute, Hinxton, UK. ²Cambridge University Hospitals NHS Foundation Trust, Cambridge, UK. ³Research Department of Pathology, University College London, London, UK. ⁴Department of Paediatrics, University of Cambridge, Cambridge, UK. ⁵Cancer Research UK Clinical Trials Unit, University of Birmingham, Birmingham, UK. ⁶Great Ormond Street Hospital for Children NHS Foundation Trust, London, UK. ⁷European Molecular Biology Laboratory, European Bioinformatics Institute (EMBL-EBI), Cambridge, UK. ⁸UCL Great Ormond Street Institute of Child Health, London, UK. ⁹Broad Institute of MIT and Harvard, Cambridge, MA, USA. ¹⁰Department of Histopathology, Royal National Orthopaedic Hospital NHS Trust, Middlesex, UK. ¹¹These authors contributed equally: Thomas R. W. Oliver, Andrew R. J. Lawson, Henry Lee-Six. ¹²These authors jointly supervised this work: Kieren Allinson, Inigo Martincorena, Thomas S. Jacques, Sam Behjati. ✉e-mail: kallinson@nhs.net; im3@sanger.ac.uk; t.jacques@ucl.ac.uk; sb31@sanger.ac.uk

## Methods

### Sample collection

This study complies with all relevant ethical regulations. Written and informed consent was given for all samples. The study of the discovery cohort of three children was approved by National Health Service (NHS) research ethics committees (PD50297−HRA East Midlands Derby REC, 08/H0405/22+5; PD51122 and PD51123−London Brent REC, 16/LO/0960). Each autopsy was undertaken at the child's local neuropathology unit, with parental consent, 1–7 days following death. Samples were snap-frozen at the point of sampling, with adjacent brain tissue taken for immediate formalin fixation and processing in the local diagnostic laboratory. A full list of the samples taken is provided in Supplementary Table 1.

The study of the validation cohort of adults with neurofibromatosis type 1 was approved by NHS research ethics committees (20/YH/0088, IRAS 272816, NHS Yorkshire and the Humber−Leeds East Research Ethics Committee). Patients with a diagnosis of neurofibromatosis type 1 who were undergoing resection for sarcoma consented to the use in research of normal tissue removed as part of the resection but distant from the lesion or of blood samples. Solid tissue samples were immediately frozen in liquid nitrogen. Blood samples were centrifuged, and plasma and cellular fractions were separated before freezing.

### Statistics and reproducibility

The study was designed in two phases. In the first phase, a discovery cohort of three children, of whom one had neurofibromatosis type 1 and two did not, was investigated. In the second phase, a validation cohort of ten adults, all of whom had neurofibromatosis type 1, was investigated.

The sample size in each case was determined by tissue availability. No statistical method was used to determine sample size. Lethal high-grade midline gliomas are rare in children−we studied all cases at collaborating centers over the study period (of approximately 2 years) in which consent for research was given. Of these, only one had neurofibromatosis type 1. For our validation cohort, all cases of patients with neurofibromatosis type 1 and tissue available at our collaborating center were studied. One patient from the validation cohort was excluded as no NF1 germline variant was identified. The experiments were not randomized, and the investigators were not blinded to whether patients had neurofibromatosis type 1 during the experiments and outcome assessment.

### Preparation of samples for sequencing: tissue processing and DNA extraction

A subset of the bulk samples in the discovery cohort underwent bulk DNA extraction using either the DNeasy Blood & Tissue Kit (Qiagen), AllPrep DNA/RNA Mini Kit (Qiagen) or the Gentra Puregene Blood Kit (Qiagen). The choice of samples to undergo bulk DNA extraction was guided by a prior understanding of the clonal architecture of the tissue. For example, intestinal biopsies were not subject to bulk DNA extraction. This is because their clonality meant that pseudo-single-cell genome readouts, rather than a single polyclonal amalgamation, could be achieved by microdissection of individual crypts instead[2]. Similarly, bulk DNA extraction was not performed for samples taken from the interface of the tumor and normal as it was hoped microdissection would better isolate the tumor and normal tissue compartments.

The remaining tissue from solid organ biopsies in the discovery cohort was fixed in PAXgene (PreAnalytiX), according to the manufacturer's instructions, and processed in preparation for laser capture microdissection using an established protocol[15]. To ensure correct feature labeling of nervous system structures during microdissection, reference slides were generated using 4-micron-thick sections mounted on SuperFrost Plus slides (VWR International) and reviewed by the neuropathologist who performed the autopsy. The sections

subject to microdissection were 16 microns thick and mounted on polyethylene naphthalate membrane slides (Leica Microsystems). The microdissected tissue then underwent lysis and further DNA extraction[15]. For duplex sequencing of samples from the first three children, curls of paraffin-embedded tissues were deparaffinized with xylene and ethanol 100% washes, followed by lysis with Arcturus PicoPure (Thermo Fisher Scientific) and DNA extraction using the DNA Micro Kit (Qiagen) according to the manufacturer's protocol save for a double elution and the use of EB as an elution buffer rather than AE.

DNA was extracted from solid tissues from the adult cohort using the DNeasy Blood & Tissue Kit (Qiagen) and from blood using the Gentra Puregene Blood Kit (Qiagen).

### Preparation of samples for sequencing: library preparation for WGS and WES

In the discovery cohort, the NEBNext Ultra II DNA Library Prep Kit (New England Biolabs) was used for the preparation of DNA extracted from the bulk samples, while the protocol for microdissected tissue used the NEBNext Ultra II DNA FS Library Prep Kit (New England Biolabs) instead. For the microdissected libraries subject to WES, the SureSelect Human All Exon V5 bait set (Agilent Technologies) was used. A full list of the successfully generated whole-genome and whole-exome sequences can be found in Supplementary Tables 3 and 4, respectively.

### Preparation of samples for sequencing: library preparation for duplex sequencing of the NF1 gene

Libraries were prepared using a version of the protocol for nanorate sequencing[23] that has been adapted to be compatible with targeted sequencing[24]. DNA was sheared to an average size of 450 bp by focused ultrasonication using the Covaris 644 LE220 instrument (Covaris) in 120 μl. It was purified using a 0.8× soluble-phase reversible immobilization (SPRI) bead ratio and eluted in 30 μl nuclease-free water (NFW). DNA fragments were blunted in a final reaction volume of 30 μl, including 3 μl (10×) mung bean buffer (Takara Bio, 2420A), 0.125 μl mung bean nuclease (Takara Bio, 2420A), 1.875 μl of NFW and 25 μl DNA. The reaction was incubated at 37 °C for 10 min with the lid tracking 5 °C above. Samples were purified using 2.5× SPRI beads and eluted in 15 μl NFW. In total, 10 μl was used as input into an A-tailing reaction, containing 1.5 μl T4 DNA ligase buffer (New England Biolabs, B0202S), 1.5 μl (1 mM) dATP/ddBTP (New England Biolabs, N0440S; GE HealthCare, 27204501), 1.5 μl Klenow fragment (3′−5′ exo-; New England Biolabs, M0212L) and 0.5 μl T4 polynucleotide kinase (New England Biolabs, M0201). The reaction was incubated at 37 °C for 30 min with the lid tracking 15 °C above. The whole sample of 15 μl was taken into the ligation reaction mix, which consisted of 30 μl Ultra II Ligation MM (New England Biolabs, E7595L), 1 μl Ultra II ligation enhancer (New England Biolabs, E7595L), 1.25 μl xGen Duplex Seq Adapters (Integrated DNA Technologies, 1080799) and 12.75 μl NFW. The reaction was incubated at 20 °C for 20 min, with the lid temperature off. Ligated DNA was cleaned up using SPRI beads and eluted in 40 μl NFW.

DNA was quantified by qPCR using a KAPA library quantification kit (Kapa Biosystems, KK4835). The supplied primer premix was first added to the supplied KAPA SYBR FAST master mix. In addition, 20 μl of 100 μM NanoqPCR1 primer (HPLC, 5′-ACACTCTTTCCCTACACGAC-3′) and 20 μl of 100 μM NanoqPCR2 primer (HPLC, 5′-GTGACTGGAGTTCAGACGTG-3′) were added to the KAPA SYBR FAST master mix. Samples were diluted 1:500 using NFW, and reactions were set up in a 10 μl reaction volume (6 μl master mix, 2 μl sample/standard and 2 μl water) in a 384-well plate. Samples were run on the Roche 480 LightCycler and analyzed using absolute quantification (second derivative maximum method) with the high-sensitivity algorithm. The concentration (nM (fmol μl$^{-1}$)) was determined as follows: (mean of sample concentration × dilution factor (500) × 452/573/1,000) × adjustment factor (1.5), where 452 represents

the size of the standard in bp, 573 is the proxy for the average fragment length of the library in bp and 1,000 is a unit conversion factor. Samples were diluted to the desired fmol amount in 25 µl using NFW.

Libraries were subsequently PCR-amplified in a 50-µl reaction volume comprising 25 µl of sample, 25 µl NEBNext Ultra II Q5 Master Mix and a unique dual index containing PCR primers (dried). The reaction was cycled as follows: step 1, 98 °C 30 s; step 2, 98 °C 10 s; step 3, 65 °C 75 s; step 4, return to step 2 (13 times); step 5, 65 °C for 5 min; step 6, hold at 4 °C. The number of PCR cycles is dependent on the input (Supplementary Table 18). The PCR product was subsequently cleaned up using two consecutive 0.7× AMPure XP clean-ups. Each sample was quantified using the AccuClear Ultra High Sensitivity dsDNA Quantification kit (Biotium). Hybrid capture was performed using TWIST hybe reagents. Samples were pooled for hybridization with 1–4 µg of PCR-amplified material per capture reaction.

### DNA sequencing

All DNA sequences were generated on the Illumina NovaSeq sequencing platform, generating paired-end 150 bp sequences. Sequences were aligned to the GRCh38 human reference genome using the Burrows–Wheeler Aligner-MEM[32]. Details on the assessment of DNA sequencing quality and sample-to-sample concordance may be found in the Supplementary Note.

### Variant calling and filtering

A detailed explanation of variant calling and filtering may be found in the Supplementary Note. In brief, for WGS and WES, substitutions were called using CaVEMan algorithm (v.1.15.1)[33], small InDels with Pindel algorithm (cgpPindel v.3.5.0)[34], copy number with both Battenberg (cgpBattenberg v.3.5.3)[35] and ASCAT (AscatNGS v.4.3.2)[36] and structural variants with GRIDSS (v.2.9.4)[37]. For duplex sequencing, mutations were detected by considering only mutations that were supported by reads from both strands that were not called in the normal sample[23,24]. After filtering, mutations were analyzed using the package dNdScv[17] (v.0.0.1.0). Code for this analysis can be found at https://github.com/trwo/nf1_second_hit_normal_tissues.

### Driver mutation identification and annotation (WGS and WES): substitutions and InDels

For substitutions and InDels, all mutations resulting in protein-coding changes in genes reported in the COSMIC (v.94) cancer gene census were initially considered[38]. Driver mutation status was assessed before the application of the exact binomial filter (which determines germline status—see above). This circumvented the risk that true driver mutations might be eliminated at a subsequent filtering step, for example, germline driver events. The following two classes of mutations were considered to be candidate drivers: first, missense substitutions or in-frame InDels occurring at hotspots in dominant-acting genes, and second, mutations in recessive-acting cancer genes predicted to result in loss of function, such as nonsense, frameshift or essential splice site variants. Candidate mutations were assigned to tiers, according to their likelihood of acting as a driver. Tier 1 substitution/InDel drivers occurred within genes that were recurrently mutated in a recent meta-analysis of over 1,000 pediatric high-grade gliomas[18]. This list of genes included *ACVR1, ASXL1, ATM, ATRX, BCOR, BRAF, CCND2, CDK4, CDK6, CDKN2A, CDKN2B, EGFR, FGFR1, H3F3A, HIST1H3B, HIST1H3C, HIST2H3C, ID2, KDM6B, KDR, KIT, KRAS, MET, MYC, MYCN, NF1, NTRK1, NTRK2, NTRK3, PDGFRA, PIK3CA, PIK3R1, PPM1D, PTEN, RB1, SETD2, TERT, TOP3A* and *TP53*. Tier 2 mutations occurred in other supposed cancer genes from the COSMIC (v.94) cancer gene census list.

Mutations in *NF1* itself were considered differently. Inactivating mutations were considered to be probable drivers. Although *NF1* is a recessive cancer gene, it does have residues that are mutated more frequently. Missense mutations that occurred in such loci, defined as >4 mutations of a given residue in COSMIC, were considered as probable

driver mutations, and further support for their functional effect was sought from the literature and from predictors of mutational effect[39].

### Driver mutation identification and annotation (WGS and WES): copy number changes

Copy number changes were determined to be driver events according to sample ploidy, the genes found on each segment and the segment length. Oncogenes were considered to be amplified if their total copy number was ≥5 when ploidy was <2.7 or ≥9 when ploidy was ≥2.7. For tumor suppressor genes, the total copy number had to equal 0 for <2.7 ploidy and ≤ (ploidy − 2.7) when ploidy was ≥2.7. Copy number aberrations passing these criteria were then annotated as putative tier 1 driver mutations if the oncogene(s) or tumor suppressor gene(s) they contained were found in the list of genes above, the segment width was ≤10 Mb wide and this was a recognized oncogenic event for that gene. For example, copy number changes in genes that mediate oncogenesis via fusion events alone were not considered tier 1 drivers. Tier 2 drivers did not meet the criteria outlined for tier 1 variants but had to be found on segments ≤1 Mb wide.

### Driver mutation identification and annotation (WGS and WES): structural variants

For a structural variant, independent of copy number state, to be annotated as a driver, it had to either form a fusion gene recognized to be oncogenic, truncate the gene footprint of a tumor suppressor gene or activate an oncogene through intragenic deletion (for example, *PDGFRA*). Once again, tier 1 events occurred in the list of genes used for other variant classes, whereas tier 2 events were plausible drivers that fell outside of these.

### Testing for recent clonal expansions associated with *NF1* null status

A linear mixed effects model comparing the mutation burden derived from WGS of *NF1* null versus *NF1*-heterozygous histologically normal CNS biopsies/microbiopsies was fitted in R using the package nlme. *NF1* null status, whether the sample was derived from bulk sequencing or laser capture microdissection, and coverage were included as fixed effects. The piece of tissue from which the sample was derived was used as a random effect (that is, two microbiopsies from the same piece of tissue should be correlated with one another). Although *NF1* null status was associated with a statistically significant effect, the effect size was only of seven additional mutations. Given that the postnatal somatic mutation rate of most tissues is 10–50 mutations per year (including a rate in glia of 27 substitutions per year[40] and a rate in neurons of 17 substitutions per year[23,40]) and the prenatal rate is usually higher, a recent clonal expansion should result in a mutation burden on the order of 100 mutations even in a child; we, therefore, concluded that *NF1* null status was incompatible with a recent clonal expansion.

### Detecting independent *NF1* null clones in normal tissue: WGS and WES

Driver events within *NF1* were initially identified in the same manner as all other driver mutations. This included a germline essential splice site *NF1* mutation within PD51122 that accounted for their neurofibromatosis type 1. Second loss of function *NF1* mutations in this case were assumed to render the affected cells '*NF1* null'.

Evidence for any *NF1* driver point mutation that had been identified in the child with neurofibromatosis type 1 was sought in the remaining two cases in the discovery cohort. Similar to the substitution filtering, this provided an approximation of the base sequencing error rate, above which we could determine an *NF1* point mutation to be truly present in the index case. We performed a one-sided Fisher's exact test using the summed variant and total read depth from the two children without *NF1* mutation against those observed in each microdissection and bulk biopsy from the child with neurofibromatosis type 1. After a

multiple hypothesis testing correction (Benjamini–Hochberg method), the *NF1* point mutation was considered present in a sample if $q < 0.01$. All *NF1* driver point mutations that were identified were found in at least one sample without any detectable tumor involvement. No copy number aberrations or structural variants involving *NF1* were detected in normal tissues using standard variant calling.

To increase our sensitivity to detect LOH events in the normal tissue of the child with neurofibromatosis type 1, we phased SNPs to each gene allele. The child's tumor had LOH of the entirety of chromosome 17, leaving only copies of the allele bearing the germline *NF1* mutation. We could phase the heterozygous SNPs identified in the deeply sequenced blood sample (PD51122q) on chromosome 17 according to which allele had the greatest allele frequency in one of the purest bulk tumor samples (PD51122m). These phased SNPs were then profiled in all remaining samples. Only SNPs with ≥10× coverage in a sample were kept for its downstream analysis, as few SNPs in noncoding regions would be captured by WES.

To identify samples with possible independent LOH events inactivating *NF1* in both the WGS and WES data, a two-sided exact binomial test was performed. In this test, the number of trials was the sum of total coverage across the heterozygous SNPs found across the gene. The number of successes was the sum of the depth of the alleles that were only found in the tumor. The hypothesized probability of success was the expected aggregate allele fraction. The *NF1* locus was $2 + 0$ in the tumor, while a normal cell would have one copy of each parental allele. The aggregate allele fraction therefore would equal tumor purity + ((1 − tumor purity) × 0.5). *P* values underwent multiple hypothesis corrections using the Benjamini–Hochberg method. To ensure confidence that we were truly detecting these in normal tissue, only samples where $q < 0.01$, the median coverage was ≥30× and tumor purity was <1% were considered to possess a copy number change to the *NF1* locus that could not be explained by tumor infiltration alone.

Two samples that had significant shifts in the proportion of each *NF1* allele unusually favored the wild-type allele (PD51122s_lo0012 and PD51122u_lo0009). These microdissections of non-neuroectodermal origin are interpreted as containing clones with LOH events that lost the mutant *NF1* allele.

The *NF1* locus had not undergone LOH in the other two children, meaning that a similar analysis could not be performed.

### Assessment of the genetic relationship between tissues that shared a somatic *NF1* variant in PD51122

For a variant to be found in two tissues, either it must have been acquired from a shared ancestor or developed independently. Assuming a comparable rate of mutation acquisition, tissues with a more recent common ancestor will share a greater number of mutations than those that are more distantly related. To determine whether the tissues carrying the same somatic *NF1* mutation were uniquely related, we first needed control to determine how related two tissues would be by chance in this child.

To construct our control data, we used the normal tissues (<1% estimated tumor contamination) without evidence of a second *NF1* hit. Separate comparisons of normal CNS versus normal CNS and normal CNS versus normal mesoderm were made to account for differences in the genetic architecture between germ layers and tissues. A mutation was determined to be shared between tissues if it was identified in both using a Shearwater-like approach (Supplementary Note), rather than relying on the calls from the variant caller alone. This improved our sensitivity for detecting low VAF variants and mitigated some of the risk that true shared variants would not be called or erroneously filtered in one sample by our pipeline. All samples in each group were then iteratively compared to the others.

The pairwise comparison was then repeated for tissues that shared a somatic *NF1* variant, and the mean number of shared substitutions per pair was calculated for each mutation (test data). The same number of pairwise comparisons for each mutation were then drawn from the control data at random, without replacement, and the mean was calculated. This was repeated 1,000 times. The *P* value was determined by the number of draws where the control data mean was greater than that observed in the test data (one-sided test).

### Reporting summary
Further information on research design is available in the Nature Portfolio Reporting Summary linked to this article.

### Data availability
WGS and targeted sequencing data are deposited in the European Genome–Phenome Archive (https://www.ebi.ac.uk/ega/) with accession ID EGAD00001015398. Mutation calls are available in Supplementary Tables 1–18. The complete catalog of substitutions identified by WGS has been deposited on Mendeley and can be accessed at https://doi.org/10.17632/hfv45sg3c5.1 (ref. 16).

### Code availability
Custom R scripts used to analyze the data can be found at https://github.com/trwo/nf1_second_hit_normal_tissues (ref. 41).

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

### Acknowledgements
This study was funded by the Wellcome Trust (institutional grant; personal fellowships to T.R.W.O. and S.B.; grants 206194, 108413/A/15/D and 223135/Z/21/Z). T.S.J. is grateful to the Brain Tumour Charity (including the Everest Centre for Low-Grade Paediatric Brain Tumours (GN-000382 and GN-000707) and the INSTINCT program), Great Ormond Street Hospital Children's Charity, Children with Cancer UK, the Olivia Hodson Cancer Fund, Cancer Research UK and the National Institute for Health Research for funding. This research was supported by the National Institute for Health and Care Research (NIHR) Great Ormond Street Hospital Biomedical Research Centre and the NIHR Biomedical Research Centre at The Royal Marsden and the Institute for Cancer Research. Additional funding was received from The Royal National Orthopaedic Research and Development Department (to A.M.F.) and The Bone Cancer Research Trust (to A.M.F.).

We thank the Children's Cancer and Leukaemia Group (CCLG) Tissue Bank, the CCLG centers and the Experimental Cancer Medicine Centres Paediatric Network for the collection and provision of tissue samples (project 2016 BS 05). The CCLG Tissue Bank is funded by Cancer Research UK and CCLG. A.M.F. is also separately supported by the National Institute for Health Research, Sarcoma UK, the UCLH Biomedical Research Centre and the UCL Experimental Cancer Centre. Funding from these institutions supported the work of the Biobank where the samples from the adult cohort were stored. H.L.-S. was supported by an NIHR Academic Clinical Fellowship and a Junior Research Fellowship from Trinity College, Cambridge, UK. The views expressed are those of the author(s) and not necessarily those of the NHS, the NIHR or the Department of Health and Social Care. We thank the clinical teams of Cambridge University Hospitals and Great Ormond Street Hospital, including the mortuary staff. V. Lee, L. Ward and O. Ogunbiyi from Great Ormond Street Hospital and A. Whyte from Addenbrooke's Hospital helped facilitate the collection and transfer of samples for which we are grateful. We thank G. Caravagna (University of Trieste) for his assistance with copy number calling quality control and A. Maartens (science writer at the Wellcome Sanger Institute) for his critical review of the manuscript. We are indebted to the families who participated in this research.

## Author contributions

S.B. and T.R.W.O. designed the experiment and wrote the manuscript. T.S.J. and K. Allinson conducted the autopsies and provided neuropathological histology expertise. T.R.W.O. performed microdissection, with laboratory support provided by Y.H. and P.A.N., and A.T. provided further technical support. T.R.W.O., A.R.J.L. and H.L.-S. analyzed data, with the assistance of R.S., M.D.Y., T.H.H.C., H.J., T.M.B., S.V.L. and I.C.-C. M.D.Y. provided statistical expertise. G.A.A.B., K. Aquilina, D.H., M.J. and T.D.T. provided clinical expertise and contributed to discussions. U.L. provided radiological expertise. F.A.J. and A.M.F. provided pathological expertise. S.B., T.S.J., I.M. and K. Allinson directed the study.

## Competing interests

I.M. is a cofounder and consultant of Quotient Therapeutics. D.H. provides consultancy to AstraZeneca/MedImmune, Alexion Pharmaceuticals, Bayer, Biodexa, Roche/Genentech and Novartis, as well as expert testimony to AstraZeneca and Novartis, and his expenses are covered by Alexion Pharmaceuticals, Boehringer Ingelheim, Roche/Genentech and Novartis. The other authors declare no competing interests.

## Additional information

**Extended data** is available for this paper at https://doi.org/10.1038/s41588-025-02097-2.

**Correspondence and requests for materials** should be addressed to Kieren Allinson, Inigo Martincorena, Thomas S. Jacques or Sam Behjati.

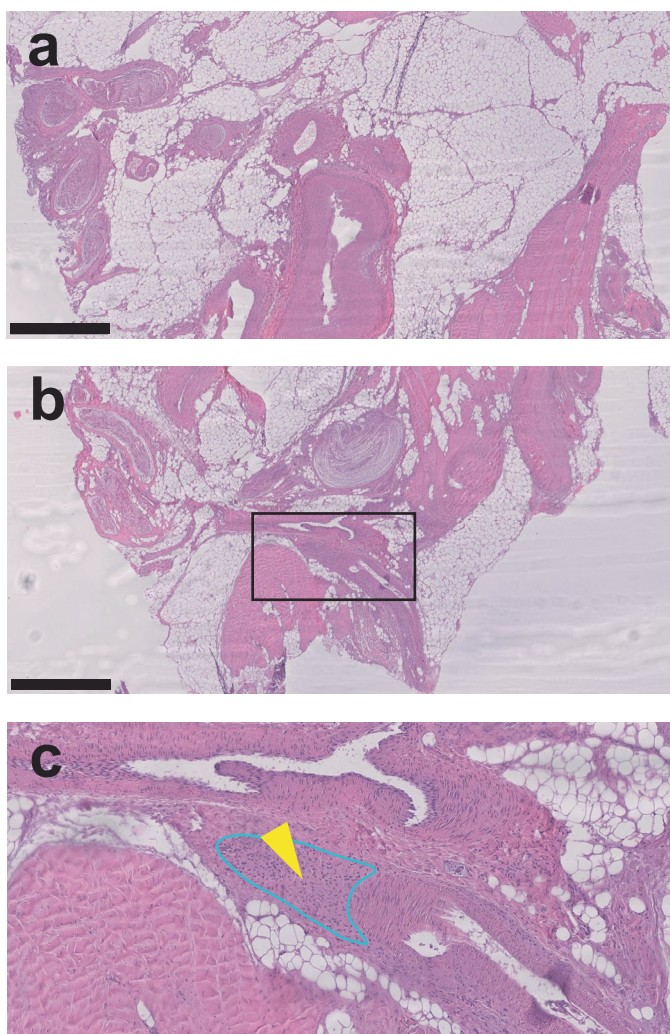

**Extended Data Fig. 1 | Subcutaneous ankle lesion in the child with neurofibromatosis type 1. a,b,** Representative histological appearances of the tissue are shown. The tissue comprises adipose tissue, bands of fibrous tissue, a Pacinian corpuscle, thick nerve trunks and ganglia. **c,** The light blue outline with a yellow arrowhead indicates the region microdissected that yielded a biallelic loss of *NF1* (PD51122ac_lo0013). It comprises bland spindled cells, embedded in fibrous and fibrillary stroma, that surround a blood vessel. The scale bars represent 1 mm, 1 mm and 250 μm, respectively.

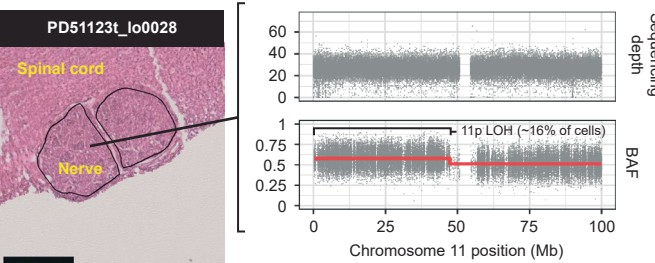

**Extended Data Fig. 2 | Mosaic uniparental disomy of chromosome 11p in a nerve from case PD51123.** Left, a histological image of the slide from which the affected sample was microdissected. Scale bar represents 250 μm. Right, the coverage of chromosome 11 SNPs and their B-allele fraction (BAF). Gridlines are present for matching to coordinates. The B-allele here is the one inferred to be derived from the major allele within predicted haplotype blocks (red line). The BAF split was not detected by ascatPCA but detected on manual review and the output here is from Battenberg.

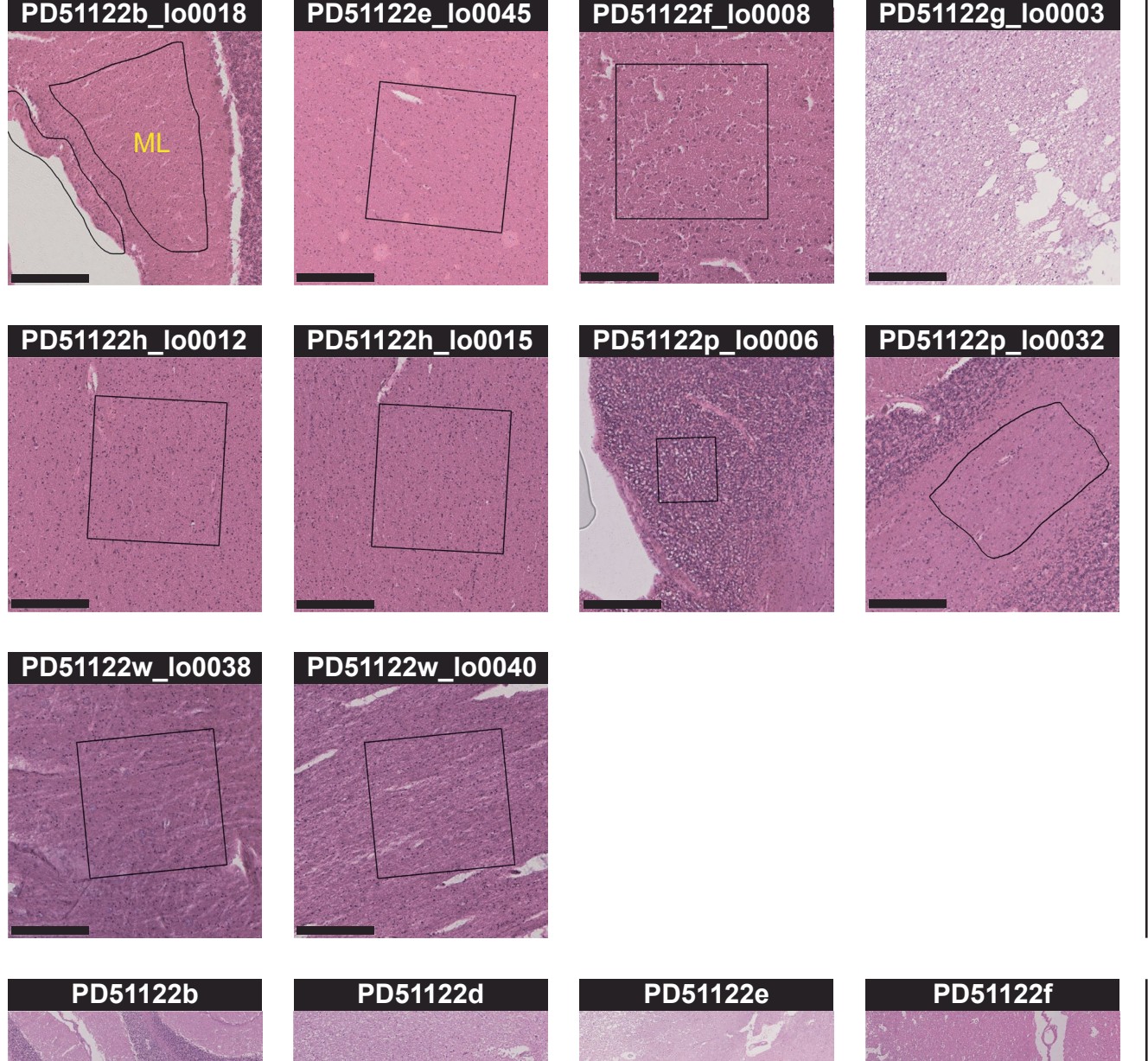

**Extended Data Fig. 3 | Histological images of normal tissues with independent *NF1* null clones.** Where the clone was detected in a microdissected sample, the exact section and area sequenced are shown. For clones found in bulk samples, a representative image from the reference slide is provided. The microdissected and reference sections are 16 μm and 4 μm thick. PD51122g_lo0003 was taken from a section that had dried before slide scanning and microdissection, resulting in a suboptimal image. Consequently, the reference slide was used here to provide an overview of the approximate area microdissected. The black outlines on images represent the perimeter of the microdissection. PD51122b_lo0018 is a region of the molecular layer (ML), taken from the cerebellum. The scale bars for the images of microdissected samples represent 250 μm. The scale bars for PD51122b, PD51122e and PD51122f are 1 mm and for PD51122d 500 μm.

## Coronal section of brain

## Cerebellum

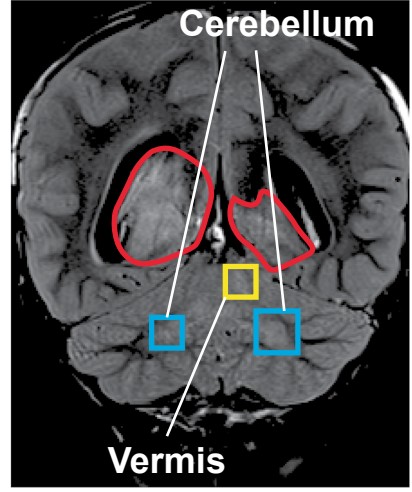

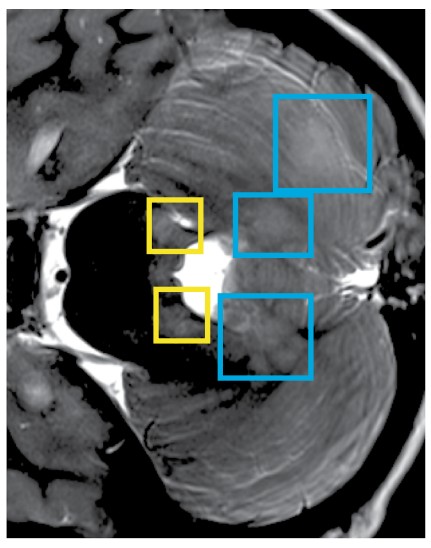

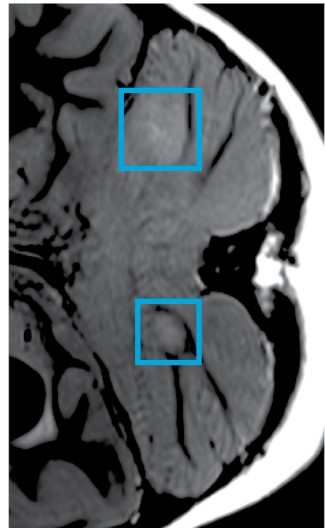

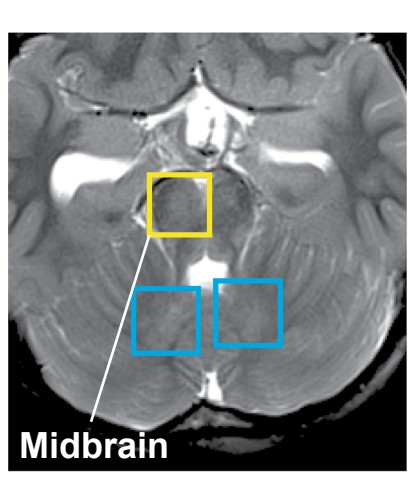

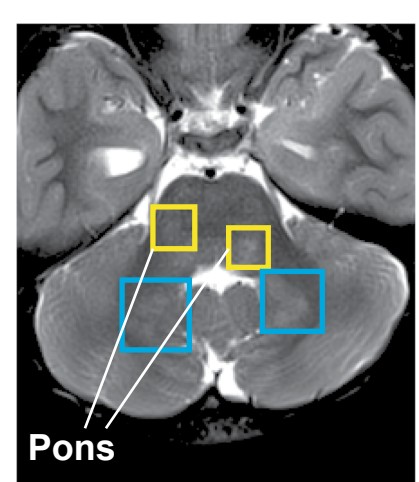

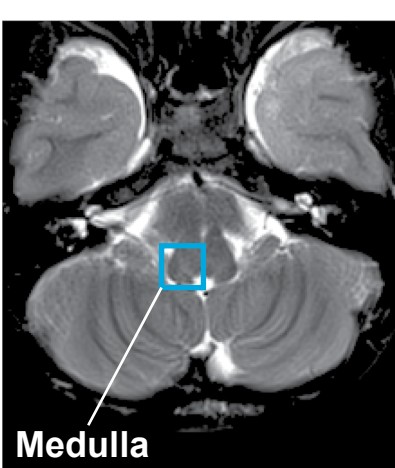

## Brainstem & surrounding structures

**Extended Data Fig. 4 | Brain MRI at diagnosis from the child with neurofibromatosis type 1.** FLAIR (top row, left and right images) and axial T2-weighted images (all other images; top row, middle and right images are rotated 90° anticlockwise) show typical focal areas of signal intensity (FASI) in the white matter and cerebellar cortex (blue boxes). The presence of tumor (red outline) makes it challenging in some areas to distinguish tumor from FASI (yellow boxes). No convincing FASI was found in the cerebral cortex or spinal cord, suggesting that no close correlation of the *NF1* null clones and FASI could be made. Note that there is hyperintensity of both hippocampi (bottom row, left image).

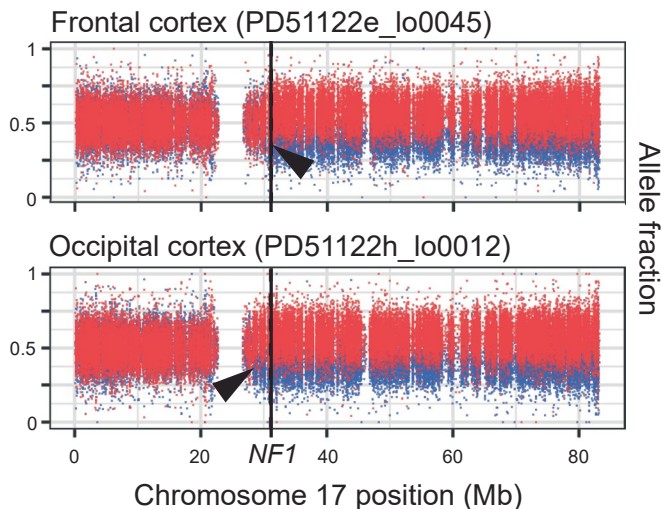

**Extended Data Fig. 5 | Allele fraction plots for the normal tissues in the child with neurofibromatosis type 1 with the largest *NF1* LOH events.** Each dot represents a heterozygous SNP that is phased to either the copy of chromosome 17 bearing the germline *NF1* mutation (red) or the wild-type allele (blue). Black arrowheads indicate the approximate location of the breakpoint in each sample.

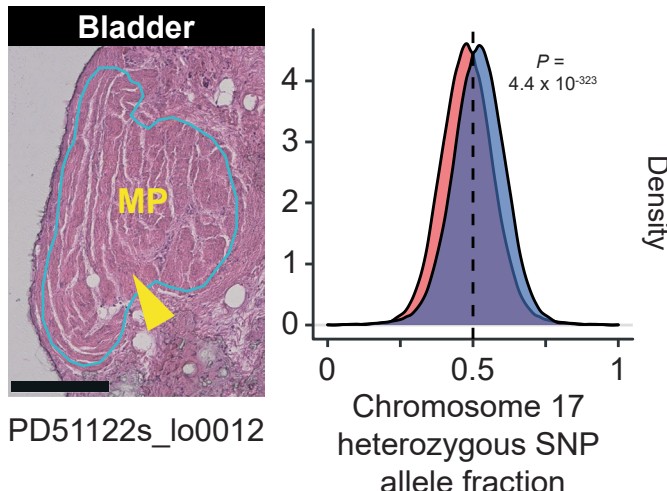

PD51122s_lo0012

**Extended Data Fig. 6 | Loss of the germline mutated *NF1* allele in bladder muscle.** Left, a histological image of the tissue microdissected to generate PD51122s_lo0012. Right, a density plot for the two alleles at heterozygous SNP sites was found across chromosome 17. A two-sided, exact binomial test is applied between the observed chromosome 17 allele fraction bearing the germline mutant *NF1* (red) and the expected fraction (dashed line; Methods). The scale bar represents 250 μm. MP, muscularis propria.

Thomas Jaques
Inigo Martincorena
Kieren Allinson

# Reporting Summary

## Statistics

For all statistical analyses, confirm that the following items are present in the figure legend, table legend, main text, or Methods section.

| n/a | Confirmed | |
|---|---|---|
| ☐ | ☒ | The exact sample size (*n*) for each experimental group/condition, given as a discrete number and unit of measurement |
| ☐ | ☒ | A statement on whether measurements were taken from distinct samples or whether the same sample was measured repeatedly |
| ☐ | ☒ | The statistical test(s) used AND whether they are one- or two-sided<br>*Only common tests should be described solely by name; describe more complex techniques in the Methods section.* |
| ☐ | ☒ | A description of all covariates tested |
| ☐ | ☒ | A description of any assumptions or corrections, such as tests of normality and adjustment for multiple comparisons |
| ☐ | ☒ | A full description of the statistical parameters including central tendency (e.g. means) or other basic estimates (e.g. regression coefficient) AND variation (e.g. standard deviation) or associated estimates of uncertainty (e.g. confidence intervals) |
| ☐ | ☒ | For null hypothesis testing, the test statistic (e.g. *F*, *t*, *r*) with confidence intervals, effect sizes, degrees of freedom and *P* value noted<br>*Give P values as exact values whenever suitable.* |
| ☐ | ☒ | For Bayesian analysis, information on the choice of priors and Markov chain Monte Carlo settings |
| ☒ | ☐ | For hierarchical and complex designs, identification of the appropriate level for tests and full reporting of outcomes |
| ☐ | ☒ | Estimates of effect sizes (e.g. Cohen's *d*, Pearson's *r*), indicating how they were calculated |

*Our web collection on statistics for biologists contains articles on many of the points above.*

## Software and code

Policy information about availability of computer code

| Data collection | No software was used for data collection. |
|---|---|
| Data analysis | Publicly available code is referenced in the manuscript (please see Methods and Supplementary Note). Software versions used were: Picard (version 2.26.10) ,Conpair (version 0.2), CaVEMan algorithm (version 1.15.1), vafCorrect (https://github.com/cancerit/vafCorrect, version 5.6.0), cgpPindel version 3.5.0, Battenberg (cgpBattenberg version 3.5.3), ascatPCA (https://github.com/hj6-sanger/ascatPCA), CNAqc (version 1.0.0), GRIDSS (version 2.9.4), GATK HaplotypeCaller (version 4.2.4.1), BCFTools (version 1.9), dNdScv (version 0.0.1.0), DPClust (version 2.2.2)<br><br>Custom R code for bespoke analyses is available at https://github.com/trwo/nf1_second_hit_normal_tissues. |

For manuscripts utilizing custom algorithms or software that are central to the research but not yet described in published literature, software must be made available to editors and reviewers. We strongly encourage code deposition in a community repository (e.g. GitHub). See the Nature Portfolio guidelines for submitting code & software for further information.

## Data

Policy information about availability of data

All manuscripts must include a data availability statement. This statement should provide the following information, where applicable:
- Accession codes, unique identifiers, or web links for publicly available datasets
- A description of any restrictions on data availability
- For clinical datasets or third party data, please ensure that the statement adheres to our policy

Whole-genome and targeted sequencing data are being deposited in the European Genome-Phenome Archive (EGA; https://www.ebi.ac.uk/ega/), with accession number with accession ID  EGAD00001015398. Mutation calls are available as supplementary tables or as a supplementary dataset on Mendeley data (https://doi.org/10.17632/hfv45sg3c5.1). Datasets used in the analyses include: gnomAD (version 3.1.1),  COSMIC (version 94), and GRCh38 human reference genome.

## Research involving human participants, their data, or biological material

Policy information about studies with human participants or human data. See also policy information about sex, gender (identity/presentation), and sexual orientation and race, ethnicity and racism.

| Reporting on sex and gender | Sex and gender were not considered in the design of the study, and are not reported in the analyses. |
|---|---|
| Reporting on race, ethnicity, or other socially relevant groupings | Race, ethnicity, and other social groupings were not considered in the design of the study, and are not reported in the analyses. |
| Population characteristics | As children who die of brain cancer are - fortunately - rare, and those with neurofibromatosis type 1 are a subset of those, care has been taken to preserve their anonymity. Age ranges have been provided rather than precise ages for this cohort, and the sex has not been provided, deliberately. For the same reason, the ethics of the study that recruited the adult cohort do not allow us to divulge precise patient characteristics that could allow patient identification. We therefore only identify them as adults with neurofibromatosis type 1. We studied all cases of children who died of brain tumours who had consented to post mortem studies in which tissue could be used for research purposes at our collaborating centres within the study period (of approximately two years). Of these, only one had neurofibromatosis type 1. For our validation cohort, all cases of patients with neurofibromatosis type 1 and tissue available at our collaborating centre were studied. |
| Recruitment | Please see details in the text and methods. |
| Ethics oversight | Study of the discovery cohort of three children was approved by NHS research ethics committees (PD50297 - HRA East Midlands Derby REC, 08/H0405/22+5; PD51122 & PD51123 - London Brent REC, 16/LO/0960). Study of the validation cohort of adults with neurofibromatosis type 1 was approved by NHS research ethics committees (20/YH/0088, IRAS 272816, NHS Yorkshire & The Humber - Leeds East Research Ethics Committee). |

Note that full information on the approval of the study protocol must also be provided in the manuscript.

# Field-specific reporting

Please select the one below that is the best fit for your research. If you are not sure, read the appropriate sections before making your selection.

☒ Life sciences          ☐ Behavioural & social sciences          ☐ Ecological, evolutionary & environmental sciences

For a reference copy of the document with all sections, see nature.com/documents/nr-reporting-summary-flat.pdf

# Life sciences study design

All studies must disclose on these points even when the disclosure is negative.

| Sample size | The study was designed in two phases. In the first phase, a discovery cohort of three children, of whom one had neurofibromatosis type 1 and two did not, was investigated. In the second phase, a validation cohort of ten adults, all of whom had neurofibromatosis type 1, was investigated. The sample size in each case was determined by tissue availability. No statistical method was used to determine sample size. Fortunately, only a small number of children die of brain tumours. We studied all cases of children who died of brain tumours who had consented to post mortem studies in which tissue could be used for research purposes at our collaborating centres within the study period (of approximately two years). Of these, only one had neurofibromatosis type 1. For our validation cohort, all cases of patients with neurofibromatosis type 1 and tissue available at our collaborating centre were studied.  The experiments were not randomized and the investigators were not blinded to whether patients had neurofibromatosis type 1 during experiments and outcome assessment. |
|---|---|
| Data exclusions | One patient from the validation cohort was excluded as no NF1 germline variant was identified. |
| Replication | Multiple sequencing modalities were used (WGS, WES, duplex sequencing) to validate the detection of mutations. Each individual experiment was not replicated, as each experiment uses up the tissue that has been investigated in the experiment itself. |

| Randomization | No randomization was carried out. Covariates were not controlled. This is not important in our study as our study is an exploratory analysis of the somatic mutation landscape in a very rare situation of a child with neurofibromatosis type 1 undergoing a post mortem examination. |
| Blinding | The study was not blinded. No conditions were tested, and so blinding would not be meaningful. |

# Reporting for specific materials, systems and methods

We require information from authors about some types of materials, experimental systems and methods used in many studies. Here, indicate whether each material, system or method listed is relevant to your study. If you are not sure if a list item applies to your research, read the appropriate section before selecting a response.

## Materials & experimental systems

| n/a | Involved in the study |
|---|---|
| ☒ | Antibodies |
| ☒ | Eukaryotic cell lines |
| ☒ | Palaeontology and archaeology |
| ☒ | Animals and other organisms |
| ☒ | Clinical data |
| ☒ | Dual use research of concern |
| ☒ | Plants |

## Methods

| n/a | Involved in the study |
|---|---|
| ☒ | ChIP-seq |
| ☒ | Flow cytometry |
| ☒ | MRI-based neuroimaging |

## Plants

| Seed stocks | Report on the source of all seed stocks or other plant material used. If applicable, state the seed stock centre and catalogue number. If plant specimens were collected from the field, describe the collection location, date and sampling procedures. |
| Novel plant genotypes | Describe the methods by which all novel plant genotypes were produced. This includes those generated by transgenic approaches, gene editing, chemical/radiation-based mutagenesis and hybridization. For transgenic lines, describe the transformation method, the number of independent lines analyzed and the generation upon which experiments were performed. For gene-edited lines, describe the editor used, the endogenous sequence targeted for editing, the targeting guide RNA sequence (if applicable) and how the editor was applied. |
| Authentication | Describe any authentication procedures for each seed stock used or novel genotype generated. Describe any experiments used to assess the effect of a mutation and, where applicable, how potential secondary effects (e.g. second site T-DNA insertions, mosiacism, off-target gene editing) were examined. |

