## [Peer Review File · Nature Genetics]

Cancer-independent somatic mutation of the wild-type NF1 allele in normal tissues in neurofibromatosis type 1

Corresponding Author: Professor Sam Behjati

A version of this paper was originally rejected for publication by Nature Genetics, however that decision was reconsidered after appeal by the authors.

Version 0:

Decision Letter:

20th Nov 2023

Dear Professor Behjati,

First, I am so sorry for the delay in returning this decision to you. Thank you for your patience.

Your Brief Communication entitled "Cancer-independent, second somatic NF1 mutation of normal tissues in neurofibromatosis type 1" has now been seen by 4 referees (Reviewers #3 and #4 reviewed together), whose comments are attached. While they find your work of potential interest, they have raised serious concerns which in our view are sufficiently important that they preclude publication of the work in Nature Genetics, at least in its present form.

While the referees find your work of some interest, they raise concerns about the strength of the novel conclusions that can be drawn at this stage.

Should further experimental data allow you to fully address these criticisms we would be willing to consider an appeal of our decision (unless, of course, something similar has by then been accepted at Nature Genetics or appeared elsewhere). This includes submission or publication of a portion of this work someplace else. I very much hope that you will consider appealing after revising the manuscript. I will be at the EBI on Thursday (23 November) so would be happy to chat to you about this if you're free.

The required new experiments and data include, but are not limited to those detailed here. We hope you understand that until we have read the revised manuscript in its entirety we cannot promise that it will be sent back for peer review.

If you are interested in attempting to revise this manuscript for submission to Nature Genetics in the future, please contact me to discuss a potential appeal. Otherwise, we hope that you find our referees' comments helpful when preparing your manuscript for resubmission elsewhere.

Sincerely,

Safia Danovi
Editor
Nature Genetics

Referee expertise:

Referee #1: somatic mutations, phylogenies, paediatric cancers

Referee #2: NF1

Referee #3: germline cancer susceptibility, tumorigenesis

Reviewers' Comments:

Reviewer #1:

Remarks to the Author:

In this brief communication by Oliver et al. entitled: "Cancer-independent, second somatic NF1 mutation of normal tissues in neurofibromatosis type 1" the authors challenge the common hypothesis that loss of heterozygosity (LOH) is the event that will induce tumorous lesions in people suffering from the cancer-predisposing syndrome neurofibromatosis type 1. Sequencing hundreds of genomes from phenotypically healthy tissue reveal the presence of multiple independent somatic LOH NF1 driver mutations. The data presented is solid and aligns with previous studies that describe the presence of tumor driving mutations in healthy tissue. While the data presented is solid, the extensive dataset seems to be only assessed at the NF1 locus. A further analysis in additional driver hits in the normal and tumorous tissue data would greatly enhance this manuscript and would make it a story worthy for publication in Nature Genetics.

Specific concerns:

1. In line 138, the authors state that they "cannot explain the non-transformation of NF1 null tissues through a lack of additional driver variants because biallelic NF1 mutation is often the sole driver event of NF1 associated neoplasms". Sequencing of tumors of NF patients has revealed that even tumors with the same mutation (interestingly the same Y489C mutation as described in the current study) may result in different severities for the patient (<https://pubmed.ncbi.nlm.nih.gov/17426081/>). In the current study, the authors have sequenced a significant number of cells of both tumor and healthy tissues with loss of NF1 expression. A further analysis of genetic modifiers that may explain why some cells do transform and others do not would greatly enhance the manuscript. Specifically in tissues where NF1-predisposed children often develop cancer, for instance the Y489C cells in the cerebellum. Although these data are presented in the supplementary tables, I would advise an additional (main) figure panel depicting these additional drivers in both normal and tumorous tissues.
2. In line 210, the authors state that LOH events seemed to have arisen at different developmental time points. The data supporting this statement is depicted in Figure 2e. Is this analysis performed solely using the depicted NF1 driver mutations? I assume passengers are also used to be sure that the same NF1 mutant lineage is studied (for example for the Y489C mutant lineage, which is found both in cerebellum and spleen). If no other mutations are used, especially passengers that are not directly subjected to selective pressure, how sure can you be that the same NF1 mutations cannot occur twice in the same individual? The whole point of the article is that LOH events occur quite often and there is a strong selection for this, which the Y489C mutation can provide. If passengers have been used to define a mutant lineage, how many of those were used/missed in the matching samples?
3. Related to the latter point, if a mutation happens extremely early, such as perhaps the Y489C mutation, there is a chance that no additional shared passengers can be used to obtain more evidence that it is the same mutant lineage. In that case, it is possible to calculate, with the data presented, what the chance is that the same LOH event can happen twice?
4. In Figure 2f, what is exactly the "early genetic similarity index" and how does this corroborate the developmental timing of NF1 mutations by considering body-wide mosaic variants (as described in line 127)? I would expect that mutations shared between tumor and normal happen earlier in development whereas mutations shared only between tumor biopsies later. However, if I get the plot correctly, it suggests the opposite.

Minor concerns:

- Typo: "arises" in line 120

- The paper is written in a way that suggests that this is a broader phenomenon in cancer pre-disposition syndromes. However only data regarding NF1 is discussed. Therefore, it still remains unclear what the broader consequences, on for instance other cancer predisposition syndromes is. A more in depth discussion regarding this would be of great value to the reader. Specifically, in lines 155-156 the authors suggest that their discovery represents a departure from conventional models of neoplasia in recessive cancer syndromes. Which syndromes would these include?

Reviewer #2:

Remarks to the Author:

The brief report by Oliver and colleagues presents whole genome sequencing of two children with diffuse midline glioma and one with neurofibromatosis type 1 (NF1). They use a number of bioinformatic methods to reveal multiple independent somatic NF1 driver mutations in histologically normal tissues, which they infer are lineages prone to neoplasia. They conclude that these preliminary findings may explain the tissue specificity and variable penetrance of oncogenesis in NF1.

Major:

How do the authors know that the LOH occurred in normal tissue?

What was the germline mutation in the one child with NF1?

How did the authors conclude that the Y489C was a driver mutation?

Line 91-92 and Figure 2. Café-au-lait macules are not neoplasms.

The data does not directly support a higher cancer phenotype, especially in tissues where tumors do not normally arise. For example, despite LOH potentially occurring in different brain regions, most pediatric brain tumors are restricted to the optic pathway and brainstem, particularly in children younger than 10 years of age.

Their comment about confirmation in experimental mouse models is puzzling. Some of the models employ biallelic Nf1 loss only, resulting in brain/nerve sheath tumors or bone abnormalities in select tissues despite more widespread Cre expression, while others that couple cell type-specific Nf1 loss with germline Nf1 heterozygosity reveal tumors in very specific brain regions (optic nerve, rather than cerebellum, cerebral hemispheres). These results coupled with other reports revealing differences in proliferation and differentiation that depend on the specific cell type (astrocytes and neural stem cells from different brain regions) suggest that other variables beyond bi-allelic loss underlie tumorigenesis. These observations were not considered.

Minor:

NF1 is not a disorder that predisposes to neuroectodermal tumors, as many of the tumors are not of neuroectodermal origin. This factual issue should be corrected.

It is not clear what the “spindle cell lesion” represented. More information should be provided.

Why do the authors believe that the spleen is the origin for JCML, rather than the bone marrow?

Reviewer #3:

Remarks to the Author:

Studies profiling various normal tissues now routinely identify somatic mutations commonly observed in cancer. While these are attributable to environmental exposures or endogenous mutational processes, currently, it is unclear if these mutational patterns are also observed among patients harboring inherited pathogenic variants in DNA repair genes. Here, Oliver et al., have conducted a truly impressive study profiling a vast number (~900) of normal tissue specimens from a pediatric patient with germline NF1 mutation as well as from two patients with sporadic brain tumors. They make several interesting observations that are unique to the germline patient. First, they report an acquisition of 2nd-hit (NF1 somatic mutation) among several histologically normal tissues of the germline NF1 mutated patient, a pattern that is not observed among sporadic tumors. Second, they perform histological and radiological evaluations and conclude that the NF1-null tissues are non-transformed. Third, they observe that some mutations were shared across tissues and likely arose during early development.

Overall, this is a remarkable finding, albeit consistent with our intuition that predisposed cells are waiting for the one somatic hit for clonal expansion and pre-neoplastic transformation. The sheer number of distinct tissues in which these NF1-biallelic clones were observed in this patient has important implications in understanding the NF1 syndrome mediated etiology and management. This study also has important implications towards spurring interest in looking at other germline predisposition genes (eg: BRCA1/2 or MMR genes). Interestingly, a study profiling normal tissues in Lynch Syndrome patients did not identify either a second hit in the MMR gene or the MMR phenotype (Lee et al., PMID: 35581206). It is possible that there are gene specific differences in presentation of this phenomenon.

Major Issues:

1. The NF1 mutations in the normal tissues are foundational to the discovery in this paper. It is critically important to present robust evidence supporting these mutations, especially given that the overall sequencing coverage is quite low (~28x). The VAF of some mutations reported here (in Figure 2E) is <10% suggesting that some NF1 mutations may be supported by 1-3 reads. The authors need to demonstrate clearly (in main or supplementary figures) that these are not artifacts or mutations observed in high mutational burden tumors (given that NF1 is a very long gene).

2. The authors describe that they have calculated a “locus-specific error rate” and called mutations that were confidently above this threshold. However, the methodology is unclear (lines 299-303). As written it is difficult to understand how each variant is evaluated against a locus-specific error rate. Similarly, it is particularly difficult to infer what ‘early variant genetic similarity index’ means. Moreover, it is unclear how the ‘locus-specific error rate’ is calculated (line 299 of methods).

3. The authors seem to use “transformed” and “neoplastic” synonymously and imply that non-neoplastic normal tissues must be non-transformed. However, the high VAFs for somatic NF1 mutations in many non-neoplastic normal tissues suggest strong positive selection. (It is hard to argue that a ~30% VAF NF1 clone (cerebellum) is not “transformed”). Presumably all mutant clones that are detectable from bulk WES or WGS have undergone prior positive selection. Further, by considering

neoplastic transformation as binary (i.e., “normal” and “neoplastic” in Figures 1A-B), this fails to consider intermediate stages where the “non-transformation of NF1” may be either because the clonal expansions are in very early stages of neoplastic transformation and/or are not detectable via histology or radiographically. Perhaps the authors could consider adopting language supporting the notion that these NF1 null clones may be on path of pre-neoplastic transformation.

4. The disease presentation, genomics, histopathology and treatment histories for each of the three patients is not described in the paper (line 53). Based on Table S1, one patient had tumors in six different regions of the brain. But that is not described in text or depicted in any of the figures. And what are the somatic NF1 mutations and other co-occurring driver alterations in these diseases (eg: TERT, ATRX, CDKN2A, TP53, PTEN, PIK3CA)? And how are the nearly ~450000 mutations reported in the supplementary table evaluated for driver status?

5. The germline NF1 mutation in the patient with neurofibromatosis does not appear to be specified anywhere in the main text or figures.

6. The table in Figure 1E lists the numbers of specimens collected but it isn't clear how many were from the primary tumors and which were from normal tissues. For readers not familiar with the intricate brain anatomy, it will be helpful to have a schematic showing the regions of the brain that were profiled for each patient and the numbers of LCM/bulk specimens collected in each region. Moreover, this spatial depiction is important to: (1) understand potential tumor infiltration into normal tissues profiled here, and (2) to understand the NF1 mutations that arose in normal tissues. As this study is poised to garner interest across the fields (genomics, bioinformatics and clinical oncology), it is important to make the anatomical descriptions of the disease as well as the specific regions sampled as interpretable as possible.

7. The premise for doing bulk tissue sequencing “to extend and validate findings” needs more clarity (line 75 of main text and line 13 of supplementary). How are they selected? Are they predominantly normal tissues in regions where NF1 mutations were detected? Moreover, line 76 suggests bulk tissue WES for 180 specimens were performed but Figure 1E indicates that the WES samples are LCMs

8. Overall the methods are difficult to read and understand. It would be helpful to present each section with first describing the premise of the methods.

9. Line 79 refers to low-level tumor contamination in normal tissues. The methods and interpretation are unclear. Was the original tumor sequenced deeply enough to capture the full spectrum of tumor specific truncal “driver” mutations, especially the NF1 somatic mutation? The authors should explain why that is not sufficient to identify and eliminate tumor infiltrating tissues. Moreover, did the authors evaluate contamination in only the specimens proximal to the tumor or all tissues, even those that are distal to the tumor.

10. Figure 2C. The authors interpret that there was no elevated mutational burden in NF1-null tissues. However, at least for a few of the NF1-null specimens, the overall substitution count was high (and highest among the bulk WGS specimen).

11. Figure 2E and Table S2: Mutations identified in bulk WGS should be highlighted distinctly from those that were LCM derived. For mutations observed in multiple tissues, it is likely that the proximity of the biopsied regions (eg: pons/medulla or parietal cortex/cerebellum) could contribute to a mutation observed in LCM in one tissue and bulk of the other. In fact, this appears to be the case for R304 and I679*. Both these were identified in LCM of one tissue and bulk WGS of a proximal tissue. The authors should re-evaluate these mutations and further exclude specimens that might have tissue infiltrates from neighboring regions.

12. Line 124: If Y489C occurred pre-gastrulation, wouldn't it have been more clonal in the spleen (Figure 2E) and also seen in other tissues with related origin? As this is a recurrent hotspot, could this mutation have arisen independently in these two tissues, especially if it was not observed in other brain and extra-cranial tissues?

Minor issues:

1. In line 35, consider changing “which may explain the tissue specificity” to “which may be explained by the tissue specificity”
2. Figure 2D and S5. A legend is needed to describe that the red histogram corresponds to the haplotype with the NF1-mutant allele.
3. Line 83: no description or interpretation of the mutation burden or substitution signatures were provided.
4. Line 113: loss of the germline NF1 may be by chance and is most likely unremarkable and definitely does not constitute a ‘reversion’.

Reviewer #4:

Remarks to the Author:

Oliver et al.: Cancer-independent, second somatic NF1 mutation of normal tissues in neurofibromatosis type 1

Studies profiling various normal tissues now routinely identify somatic mutations commonly observed in cancer. While these are attributable to environmental exposures or endogenous mutational processes, currently, it is unclear if these mutational patterns are also observed among patients harboring inherited pathogenic variants in DNA repair genes. Here, Oliver et al.,

have conducted a truly impressive study profiling a vast number (~900) of normal tissue specimens from a pediatric patient with germline NF1 mutation as well as from two patients with sporadic brain tumors. They make several interesting observations that are unique to the germline patient. First, they report an acquisition of 2nd-hit (NF1 somatic mutation) among several histologically normal tissues of the germline NF1 mutated patient, a pattern that is not observed among sporadic tumors. Second, they perform histological and radiological evaluations and conclude that the NF1-null tissues are non-transformed. Third, they observe that some mutations were shared across tissues and likely arose during early development.

Overall, this is a remarkable finding, albeit consistent with our intuition that predisposed cells are waiting for the one somatic hit for clonal expansion and pre-neoplastic transformation. The sheer number of distinct tissues in which these NF1-biallelic clones were observed in this patient has important implications in understanding the NF1 syndrome mediated etiology and management. This study also has important implications towards spurring interest in looking at other germline predisposition genes (eg: BRCA1/2 or MMR genes). Interestingly, a study profiling normal tissues in Lynch Syndrome patients did not identify either a second hit in the MMR gene or the MMR phenotype (Lee et al., PMID: 35581206). It is possible that there are gene specific differences in presentation of this phenomenon.

Major Issues:

1. The NF1 mutations in the normal tissues are foundational to the discovery in this paper. It is critically important to present robust evidence supporting these mutations, especially given that the overall sequencing coverage is quite low (~28x). The VAF of some mutations reported here (in Figure 2E) is <10% suggesting that some NF1 mutations may be supported by 1-3 reads. The authors need to demonstrate clearly (in main or supplementary figures) that these are not artifacts or mutations observed in high mutational burden tumors (given that NF1 is a very long gene).
2. The authors describe that they have calculated a "locus-specific error rate" and called mutations that were confidently above this threshold. However, the methodology is unclear (lines 299-303). As written it is difficult to understand how each variant is evaluated against a locus-specific error rate. Similarly, it is particularly difficult to infer what 'early variant genetic similarity index' means. Moreover, it is unclear how the 'locus-specific error rate' is calculated (line 299 of methods).
3. The authors seem to use "transformed" and "neoplastic" synonymously and imply that non-neoplastic normal tissues must be non-transformed. However, the high VAFs for somatic NF1 mutations in many non-neoplastic normal tissues suggest strong positive selection. (It is hard to argue that a ~30% VAF NF1 clone (cerebellum) is not "transformed".) Presumably all mutant clones that are detectable from bulk WES or WGS have undergone prior positive selection. Further, by considering neoplastic transformation as binary (i.e., "normal" and "neoplastic" in Figures 1A-B), this fails to consider intermediate stages where the "non-transformation of NF1" may be either because the clonal expansions are in very early stages of neoplastic transformation and/or are not detectable via histology or radiographically. Perhaps the authors could consider adopting language supporting the notion that these NF1 null clones may be on path of pre-neoplastic transformation.
4. The disease presentation, genomics, histopathology and treatment histories for each of the three patients is not described in the paper (line 53). Based on Table S1, one patient had tumors in six different regions of the brain. But that is not described in text or depicted in any of the figures. And what are the somatic NF1 mutations and other co-occurring driver alterations in these diseases (eg: TERT, ATRX, CDKN2A, TP53, PTEN, PIK3CA)? And how are the nearly ~450000 mutations reported in the supplementary table evaluated for driver status?
5. The germline NF1 mutation in the patient with neurofibromatosis does not appear to be specified anywhere in the main text or figures.
6. The table in Figure 1E lists the numbers of specimens collected but it isn't clear how many were from the primary tumors and which were from normal tissues. For readers not familiar with the intricate brain anatomy, it will be helpful to have a schematic showing the regions of the brain that were profiled for each patient and the numbers of LCM/bulk specimens collected in each region. Moreover, this spatial depiction is important to: (1) understand potential tumor infiltration into normal tissues profiled here, and (2) to understand the NF1 mutations that arose in normal tissues. As this study is poised to garner interest across the fields (genomics, bioinformatics and clinical oncology), it is important to make the anatomical descriptions of the disease as well as the specific regions sampled as interpretable as possible.
7. The premise for doing bulk tissue sequencing "to extend and validate findings" needs more clarity (line 75 of main text and line 13 of supplementary). How are they selected? Are they predominantly normal tissues in regions where NF1 mutations were detected? Moreover, line 76 suggests bulk tissue WES for 180 specimens were performed but Figure 1E indicates that the WES samples are LCMs.
8. Overall the methods are difficult to read and understand. It would be helpful to present each section with first describing the premise of the methods.
9. Line 79 refers to low-level tumor contamination in normal tissues. The methods and interpretation are unclear. Was the original tumor sequenced deeply enough to capture the full spectrum of tumor specific truncal "driver" mutations, especially the NF1 somatic mutation? The authors should explain why that is not sufficient to identify and eliminate tumor infiltrating tissues. Moreover, did the authors evaluate contamination in only the specimens proximal to the tumor or all tissues, even those that are distal to the tumor.
10. Figure 2C. The authors interpret that there was no elevated mutational burden in NF1-null tissues. However, at least for

a few of the NF1-null specimens, the overall substitution count was high (and highest among the bulk WGS specimen).

11. Figure 2E and Table S2: Mutations identified in bulk WGS should be highlighted distinctly from those that were LCM derived. For mutations observed in multiple tissues, it is likely that the proximity of the biopsied regions (eg: pons/medulla or parietal cortex/cerebellum) could contribute to a mutation observed in LCM in one tissue and bulk of the other. In fact, this appears to be the case for R304 and I679*. Both these were identified in LCM of one tissue and bulk WGS of a proximal tissue. The authors should re-evaluate these mutations and further exclude specimens that might have tissue infiltrates from neighboring regions.

12. Line 124: If Y489C occurred pre-gastrulation, wouldn't it have been more clonal in the spleen (Figure 2E) and also seen in other tissues with related origin? As this is a recurrent hotspot, could this mutation have arisen independently in these two tissues, especially if it was not observed in other brain and extra-cranial tissues?

Minor issues:

1. In line 35, consider changing "which may explain the tissue specificity" to "which may be explained by the tissue specificity"
2. Figure 2D and S5. A legend is needed to describe that the red histogram corresponds to the haplotype with the NF1-mutant allele.
3. Line 83: no description or interpretation of the mutation burden or substitution signatures were provided.
4. Line 113: loss of the germline NF1 may be by chance and is most likely unremarkable and definitely does not constitute a 'reversion'.

Version 1:

Decision Letter:

IMPORTANT: Please note the reference number: NG-BC63506R-Z Behjati. This number must be quoted whenever you communicate with us regarding this paper.

15th Jul 2024

Dear Dr Behjati,

Thank you for asking us to reconsider our decision on your manuscript "Cancer-independent, second somatic NF1 mutation of normal tissues in neurofibromatosis type 1". I have now discussed your appeal with my colleagues, and we think that you have some valid points. We therefore invite you to revise your manuscript along the lines that you propose.

When preparing a revision, please ensure that it fully complies with our editorial requirements for format and style; details can be found in the Guide to Authors on our website (<http://www.nature.com/ng/>).

Please be sure that your manuscript is accompanied by a separate letter detailing the changes you have made and your response to the points raised. At this stage we will need you to upload:

- 1) a copy of the manuscript in MS Word .docx format.
- 2) The Editorial Policy Checklist (presently missing):
<https://www.nature.com/documents/nr-editorial-policy-checklist.pdf>
- 3) The Reporting Summary (presently missing):
<https://www.nature.com/documents/nr-reporting-summary.pdf>
(Here you can read about the role of the Reporting Summary in reproducible science:
<https://www.nature.com/news/announcement-towards-greater-reproducibility-for-life-sciences-research-in-nature-1.22062>)

Please use the link below to be taken directly to the site and view and revise your manuscript:

Link Redacted

With kind wishes,

Safia Danovi, PhD
Senior Editor, Nature Genetics
ORCID: 0009-0007-7822-5479

Version 2:

Decision Letter:

7th Aug 2024

Dear Sam,

Your Letter, "Cancer-independent, second somatic NF1 mutation of normal tissues in neurofibromatosis type 1" has now been seen by 4 referees. As in the last round, Reviewers #3 and #4 reviewed together and uploaded the same report. You will see from their comments below that while they find your work of interest, some important points are raised. We are interested in the possibility of publishing your study in Nature Genetics, but would like to consider your response to these concerns in the form of a revised manuscript before we make a final decision on publication.

We therefore invite you to revise your manuscript taking into account all reviewer and editor comments. Please highlight all changes in the manuscript text file. At this stage we will need you to upload a copy of the manuscript in MS Word .docx or similar editable format.

Depending on your response, we will either assess the revisions in-house (this is our preferred option) or return to one or more of the reviewers. We'll only choose the latter if absolutely necessary.

*2) If you have not done so already please begin to revise your manuscript so that it conforms to our Letter format instructions, available

[here](http://www.nature.com/ng/authors/article_types/index.html).

*3) Include a revised version of any required Reporting Summary: <https://www.nature.com/documents/nr-reporting-summary.pdf>

Link Redacted

We hope to receive your revised manuscript within four to eight weeks. If you cannot send it within this time, please let us know.

Nature Genetics is committed to improving transparency in authorship. As part of our efforts in this direction, we are now requesting that all authors identified as 'corresponding author' on published papers create and link their Open Researcher and Contributor Identifier (ORCID) with their account on the Manuscript Tracking System (MTS), prior to acceptance. ORCID helps the scientific community achieve unambiguous attribution of all scholarly contributions. You can create and link your

ORCID from the home page of the MTS by clicking on 'Modify my Springer Nature account'. For more information please visit please visit www.springernature.com/orcid.

Sincerely,

Safia Danovi, PhD
Senior Editor, Nature Genetics
ORCID: 0009-0007-7822-5479

Referee expertise:

Reviewers' Comments:

Reviewer #1:

Remarks to the Author:

The authors have addressed all my comments. I would like to congratulate them with a wonderful paper. I have only two small points for their consideration:

- Is it possible to phase some of the 29 non-synonymous mutations found by duplex sequencing to the allele which is not affected by the germline NF1 mutations? As pointed out in the manuscript, the authors know which SNPs are located on the germline mutated allele thanks to the LOH event. Of course, not all the 29 non-synonymous mutations can probably be phased, but for the ones that can be, it would be good to validate that these are indeed located on the WT allele.

- The title of the graphs in Figure 2a is on the wrong side (right instead of left side of the graph).

Reviewer #2:

Remarks to the Author:

The brief report by Oliver and colleagues presents whole genome sequencing of two children with diffuse midline glioma and one with neurofibromatosis type 1 (NF1). They use a number of bioinformatic methods to reveal multiple independent somatic NF1 driver mutations in histologically normal tissues, which they infer are lineages prone to neoplasia. They conclude that these preliminary findings may explain the tissue specificity and variable penetrance of oncogenesis in NF1.

It is unclear why the authors conclude Y489C is a mutational hotspot. There is no clear evidence for mutational hotspots in the NF1 gene, either in the context of neurofibromatosis or in sporadic cancers.

Why is it unlikely that additional mutations would occur in tumor cells? There are descriptions of additional (rare) somatic mutations in PTEN, BRAF (K11A1549:BRAF), and FGFR1 in NF1-associated brain tumors.

Is there evidence for a founder mutational effect, where regionally related cells share the same somatic mutation?

While leukemic cells do get trapped in the spleen, this may not be their cell of origin. The sentence should be revised.

The sentence about "transformation to a frank malignancy" should be revised to reflect the fact that the majority of tumors arising in the setting of NF1 are non-malignant (benign) tumors.

Reviewer #3:

Remarks to the Author:

The revised study is now substantially improved and addresses many of my original comments. The authors have generated new data and have now quantitatively established evidence for positive selection of clones harboring somatic NF1 in normal tissues of predisposed individuals. This is a very interesting finding and supports our intuition and will be a significant addition to literature.

Some minor concerns mainly surrounding the clarity with which the results are presented remain.

1. The level of detail with which the evidence for the mutations highlighted in the study is underwhelming. Table S6 does not have the VAFs or the read coverage information. It is helpful to include the VAFs in Figure 2C. Currently, there is no way to understand the evidence supporting the genotyping VAFs shown in Figure 2E. For example, I679fs* was originally detected at 10% in only one of the non-neoplastic tissues. But in the genotyping step, the authors report detecting this mutation at >>10% VAF in 3 tissues in which the original mutation calling pipeline did not detect this event. This may be because the

locus-specific error rate is very low. But is it possible that the overall background rate of mutations is very high in those tissues in which these high VAF mutations are genotyped?

2. Methods have been substantially revised but there is a still lot missing. The authors need to describe how they performed the permutation test for Figure 2E (for example, how is a permutation test performed with just one data point for NF1 p.Y489C), or the dN/dS calculations in Figure 2F, G. For indels, the authors report using "Exonerate". But how was genotyping performed for SNVs?

3. Regarding 2E, why are there a median of 2 shared mutations between all pairs of CNS vs. CNS and CNS vs. WES samples? Isn't the expectation that there are 0 shared mutations between clonally unrelated tissues? Do the authors hypothesize that these mutations are independently arising or could they be some type of artifact?

Reviewer #4:

Remarks to the Author:

The revised study is now substantially improved and addresses many of my original comments. The authors have generated new data and have now quantitatively established evidence for positive selection of clones harboring somatic NF1 in normal tissues of predisposed individuals. This is a very interesting finding and supports our intuition and will be a significant addition to literature.

Some minor concerns mainly surrounding the clarity with which the results are presented remain.

1. The level of detail with which the evidence for the mutations highlighted in the study is underwhelming. Table S6 does not have the VAFs or the read coverage information. It is helpful to include the VAFs in Figure 2C. Currently, there is no way to understand the evidence supporting the genotyping VAFs shown in Figure 2E. For example, I679fs* was originally detected at 10% in only one of the non-neoplastic tissues. But in the genotyping step, the authors report detecting this mutation at >>10% VAF in 3 tissues in which the original mutation calling pipeline did not detect this event. This may be because the locus-specific error rate is very low. But is it possible that the overall background rate of mutations is very high in those tissues in which these high VAF mutations are genotyped?

2. Methods have been substantially revised but there is a still lot missing. The authors need to describe how they performed the permutation test for Figure 2E (for example, how is a permutation test performed with just one data point for NF1 p.Y489C), or the dN/dS calculations in Figure 2F, G. For indels, the authors report using "Exonerate". But how was genotyping performed for SNVs?

3. Regarding 2E, why are there a median of 2 shared mutations between all pairs of CNS vs. CNS and CNS vs. WES samples? Isn't the expectation that there are 0 shared mutations between clonally unrelated tissues? Do the authors hypothesize that these mutations are independently arising or could they be some type of artifact?

Version 3:

Decision Letter:

Our ref: NG-BC63506R2

25th Sep 2024

Dear Dr Behjati,

Thank you for submitting your revised manuscript "Cancer-independent, second somatic NF1 mutation of normal tissues in neurofibromatosis type 1" (NG-BC63506R2). It has now been seen by the original referees and their comments are below (again, Reviewers #3 and #4 reviewed together). The reviewers find that the paper has improved in revision, and therefore we'll be happy in principle to publish it in Nature Genetics, pending minor revisions to satisfy our editorial and formatting guidelines. Please note that we'd suggest upgrading the manuscript to a Letter as the word count is too high for a Brief Communication.

Sincerely,

Safia Danovi, PhD

Senior Editor, Nature Genetics
ORCID: 0009-0007-7822-5479

Reviewer #1 (Remarks to the Author):

The authors have addressed all my comments.

Reviewer #2 (Remarks to the Author):

The authors have done a satisfactory job addressing my comments and critiques

Reviewer #3 (Remarks to the Author):

I believe the authors have satisfactorily addressed all of our concerns and commend them on an interesting study.

Reviewer #4 (Remarks to the Author):

The authors have satisfactorily addressed all of our concerns and that we congratulate them on an interesting manuscript.

Reviewer #4 (Remarks on figshare data availability):

It appears the authors have provided sufficient data to reproduce key findings in the study.

Rebuttal for manuscript ‘Cancer-independent, second somatic *NF1* mutation of normal tissues in neurofibromatosis type 1’

We thank the Editor and Reviewers for their comments. We have considerably revised and expanded our manuscript to address the questions raised. In addition to clarifying our previous data, highlights of this revision include:

- To enable a formal statistical assessment of selection for non-synonymous variants, we performed Duplex sequencing (enabling single DNA molecule mutation calling) of the *NF1* gene in normal tissues from the three post mortem cases. These data validate our finding, showing strong statistical evidence that *NF1* is under positive selection in the child with type 1 neurofibromatosis.
- We have obtained histologically normal, uninvolved peripheral nerves, muscle, and blood samples from nine individuals with neurofibromatosis type 1 who had undergone resections for sarcomas. We subjected these to duplex sequencing of the *NF1* gene. This again shows that (non-synonymous) second *NF1* hits occur frequently in predisposed tissues.
- In addition, we have extensively revised / rewritten the manuscript and added data tables, as per the Reviewers’ suggestions.

We hope that the Reviewers would agree that these additional data provide compelling evidence of the selection of second *NF1* hits in predisposed individuals as a general principle of driver acquisition in neurofibromatosis type 1.

Referee 1 – somatic mutations, phylogenies, paediatric cancers

#	Comment	Reply
1.1	In this brief communication by Oliver et al. entitled: “Cancer-independent, second somatic NF1 mutation of normal tissues in neurofibromatosis type 1” the authors challenge the common hypothesis that loss of heterozygosity (LOH) is the event that will induce tumorous lesions in people suffering from the cancer-predisposing syndrome neurofibromatosis type 1. Sequencing hundreds of genomes from phenotypically healthy tissue reveal the presence of multiple independent somatic LOH NF1 driver mutations. The data presented is solid and aligns with previous studies that describe the presence of tumor driving mutations in healthy tissue. While the data presented is solid, the extensive dataset seems to be only assessed at the NF1	We thank the Reviewer for the kind comments and apologise for the lack of clarity. We have listed plausible driver events in any cancer gene in any tissue in a dedicated table in the main manuscript (Table 1). Leaving aside NF1 variants, we found the following plausible driver events (see Table 1 for details) in the following genes across ~800 normal tissues: 1 x CREBBP 2 x MSH6 In ~170 tumour biopsies (from three children), we found driver events in glioma-typical cancer genes, including PDGFRA, MYCN, CDK6, CDKN2A, EGFR, PIK3CA, KIT, KRAS, H3F3A, NF1, PIK3R1, ATRX, CCND2, TP53, PTEN, RB1 and KDR.

	locus. A further analysis in additional driver hits in the normal and tumorous tissue data would greatly enhance this manuscript and would make it a story worthy for publication in Nature Genetics.	Changes to manuscript: 1) Addition of Table 1 to main manuscript
1.2	Specific concerns: 1. In line 138, the authors state that they “cannot explain the non-transformation of NF1 null tissues through a lack of additional diver variants because biallelic NF1 mutation is often the sole driver event of NF1 associated neoplasms”. Sequencing of tumors of NF patients has revealed that even tumors with the same mutation (interestingly the same Y489C mutation as described in the current study) may result in different severities for the patient (https://pubmed.ncbi.nlm.nih.gov/17426081/ [pubmed.ncbi.nlm.nih.gov]). In the current study, the authors have sequenced a significant number of cells of both tumor and healthy tissues with loss of NF1 expression. A further analysis of genetic modifiers that may explain why some cells do transform and others do not would greatly enhance the manuscript. Specifically in tissues where NF1-predisposed children often develop cancer, for instance the Y489C cells in the cerebellum. Although these data are presented in the supplementary tables, I would advise an additional (main) figure panel depicting these additional drivers in both normal and tumorous tissues.	We thank the Reviewer for highlighting the lack of clarity in our initial manuscript. To address this concern, we added a data table to the main manuscript, as detailed in our response to your point above (1.1). For clarification, no further driver events were identified in the NF1 null clones seen in normal tissues. We have added an extended discussion where we speculate as to why some clones transform and others do not. Changes to manuscript: 1) Addition of Table 1 to main manuscript 2 Extended discussion on the findings that determine whether NF1 null clones transform or not (lines 203-214).
1.3	2. In line 210, the authors state that LOH events seemed to have arises at different developmental time points. The data supporting this statement is depicted in Figure 2e. Is this analysis performed solely using the depicted NF1 driver mutations? I assume passengers are also used to be sure that the same NF1 mutant lineage is studied (for example for the Y489C mutant lineage, which is found both in cerebellum and spleen). If no other mutations are used, especially	Thank you for these related comments. This is a highly pertinent point which we should have discussed in more detail. In suggesting specific embryological timing, we had also considered phylogenetic passenger mutations that corroborate the timing of each of the mutations. As the Reviewer rightly highlights, in “late” NF1 mutations (i.e. mutations that occurred late in development), there are more such corroborating mutations than in the early Y489C mutation. Whilst a quantification may be invariably superficial, as we

	passengers that are not directly subjected to selective pressure, how sure can you be that the same NF1 mutations cannot occur twice in the same individual? The whole point of the article is that LOH events occur quite often and there is a strong selection for this, which the Y489C mutation can provide. If passengers have been used to define a mutant lineage, how many of those were used/missed in the matching samples? 3. Related to the latter point, if a mutation happens extremely early, such as perhaps the Y489C mutation, there is a chance that no additional shared passengers can be used to obtain more evidence that it is the same mutant lineage. In that case, it is possible to calculate, with the data presented, what the chance is that the same LOH event can happen twice?	do not know many of the variables, we can measure the phylogenetic relatedness of each NF1 mutant tissue to the other NF1 mutant tissues which provides additional (though clearly not definitive) evidence supporting our claims of independence and sharedness. We find that tissues within the CNS that share an inactivating mutation in NF1 are statistically significantly more closely related to one another than they are to other tissues that lack the NF1 mutation, suggesting that they share a common ancestor. In contrast, the spleen and brain sample that shared the Y489C mutation were not more closely related to one another than to other tissues, which indicates that this mutation is likely to have occurred twice independently. We have now included this analysis in the manuscript (Figure 2e), and, considering the Reviewer's comments and our new Duplex sequencing data of NF1 (which shows just how common second driver events are in NF1 in this child's tissues), we have revised our interpretation of the data to acknowledge the possibility of parallel emergence of the same driver events, even when tissues share the same developmental ancestry. Changes to manuscript: 1) Addition of Figure 2e 2) Revised interpretation of the data in light of this analysis and the insights gained from the additional cases sequenced (lines 138-150).
1.4	4. In Figure 2f, what is exactly the "early genetic similarity index" and how does this corroborate the developmental timing of NF1 mutations by considering body-wide mosaic variants (as described in line 127)? I would expect that mutations shared between tumor and normal happen earlier in development whereas mutations shared only between tumor biopsies later. However, if I get the plot correctly, it suggests the opposite.	We apologise for the lack of clarity. The data shown in Figure 2f were meant to ask the question – Are all NF1 null cells derived from a common embryonic lineage? To which the answer in pairwise comparisons is "no" (hence, there is no difference in genetic proximity between NF1 null normal and NF1 heterozygous normal tissues). However, in the comparison (we performed to address points 1.2 & 1.3) of the proximity between NF1 null tissues with the same second hit VS NF1 null tissues with other different hits, we can see that CNS tissues with the same mutation are more closely related than those without. In the revised manuscript, we have removed the analysis you describe. We hope the revised manuscript with the additional data provides greater clarity than the original. Changes to manuscript: 1) Removal of the original Figure 2f

1.5	Minor concerns: - Typo: “arises” in line 120	Thank you for spotting this. Corrected.
1.6	- The paper is written in a way that suggests that this is a broader phenomenon in cancer pre-disposition syndromes. However only data regarding NF1 is discussed. Therefore, it still remains unclear what the broader consequences, on for instance other cancer predisposition syndromes is. A more in depth discussion regarding this would be of great value to the reader. Specifically, in lines 155-156 the authors suggest that their discovery represents a departure from conventional models of neoplasia in recessive cancer syndromes. Which syndromes would these include?	Encouraged by this suggestion, we have extended our thoughts and discussion. We are careful to limit our claims to neurofibromatosis type 1, but raise the possibility that it may occur more widely. We are hesitant to name other syndromes that may follow the same pattern as neurofibromatosis type 1, as those data are lacking, but highlight this as an area that requires research. Changes to manuscript: 1) Extended discussion on the implications of this observation for our model of understanding tumorigenesis in individuals with recessive cancer gene predisposition syndromes (lines 216-218).

Referee 2 –NF1

#	Comment	Reply
2.1	The brief report by Oliver and colleagues presents whole genome sequencing of two children with diffuse midline glioma and one with neurofibromatosis type 1 (NF1). They use a number of bioinformatic methods to reveal multiple independent somatic NF1 driver mutations in histologically normal tissues, which they infer are lineages prone to neoplasia. They conclude that these preliminary findings may explain the tissue specificity and variable penetrance of oncogenesis in NF1.	Thank you for appraising our manuscript.
2.2	Major: How do the authors know that the LOH occurred in normal tissue?	There are two parts to this question: how we define ‘normality’, and how we assessed LOH. To answer the first part, the tissues were macroscopically and histologically normal. In cases where we found LOH in microdissections, we can go further and say that the

		exact tissue we excised and sequenced looked normal at high power magnification. This is the standard that has been used in recent studies that looked for mutations in normal tissues [1-3]. To exclude the possibility that these LOH events reflect contamination by the glioma, we looked for any reads supporting mutations found within the tumour's trunk within all normal tissue and validated this method using simulated data (please see the Methods section) To answer the second, we assessed the allele frequency of heterozygous SNPs on chromosome 17 in every sample. As we could separate parental haploblocks (due to whole chromosome 17 LOH in the very pure tumour), we were able to definitively assign heterozygous SNPs to parental haploblocks ("phase") which makes LOH calls extraordinarily and beautifully robust. Chromosome 17 LOH occurred in anatomically distant areas (please see the new Table 1 that summarises this) and with distinct breakpoints (Supplementary Figure 5), indicating that this event has occurred multiple times independently. This would not be consistent with either tumour contamination or an LOH event in a precursor clone.  1. Lawson, A.R.J. et al. Extensive heterogeneity in somatic mutation and selection in the human bladder. Science 370, 75-82 (2020). 2. Lee-Six, H. et al. The landscape of somatic mutation in normal colorectal epithelial cells. Nature 574, 532-537 (2019). 3. Moore, L. et al. The mutational landscape of human somatic and germline cells. Nature 597, 381-386 (2021).
2.3	What was the germline mutation in the one child with NF1?	We apologise that this information was buried in the manuscript. This child has an essential splice site (c.3113+1G>A) germline mutation in NF1. Changes to manuscript: 1) We have now included this prominently in the main text (line 76).
2.4	How did the authors conclude that the Y489C was a driver mutation?	We followed four lines of evidence (below) to conclude that Y489C was a driver mutation. We have expanded our Methods section to explain this.

First, it is a mutational hotspot. In Cosmic (<https://cancer.sanger.ac.uk/cosmic/gene/analysis?ln=NF1>), Y489C has been reported to be mutated 16 times in human cancer; 14 times as Y489C, predominantly in classical *NF1* related cancers. Y489C is the most common somatic missense mutation in *NF1*.

Second, we assessed Y489C in the Clinvar database (<https://www.ncbi.nlm.nih.gov/clinvar/variation/354/#new-submission-germline>). Y489C has been reported as pathogenic in the germline, with 27 individual submissions to Clinvar supporting its pathogenicity, and a 2* rating (indicating multiple submissions that are concordant with one another).

Third, we used the Karolinska Molecular Tumour Board programme (<https://www.mtbp.org/>) for an *in silico* prediction of the structural effect of the variant. Y489C was predicted to be pathogenic.

Finally, we searched the literature for experimental evidence of the functional effect of the mutation. RNA sequencing studies have suggested that Y489C affects splicing, resulting in a new splice donor site that leads to the deletion of part of exon 13 (<https://www.nature.com/articles/s41525-021-00258-w>)

		Changes to manuscript: 1) Description of how mutations in NF1 were assessed for their driver status in the Methods section.
2.5	Line 91-92 and Figure 2. Café-au-lait macules are not neoplasms.	Thank you for raising this point. We took ‘neoplasm’ in its literal sense, as a new growth deriving from a single cell. Café-au-lait macules have been shown to be derived from expansions of single melanocytes, as they have clonal second hits in NF1 (see, e.g. Eisenbarth I, Assum G, Kaufmann D, Krone W. Evidence for the presence of the second allele of the neurofibromatosis type 1 gene in melanocytes derived from cafe au lait macules of NF1 patients. Biochem Biophys Res Commun 1997;237:138–141. or De Schepper S, Maertens O, Callens T, Naeyaert JM, Lambert J, Messiaen L. Somatic mutation analysis in NF1 cafe au lait spots reveals two NF1 hits in the melanocytes. J Invest Dermatol 2008;128:1050–1053.) To avoid confusion, however, we have altered our description of café-au-lait macules to ‘macroscopically visible clonal expansion of melanocytes’. Changes to manuscript: 1) Rephrased our description of café-au-lait macules (line 68)
2.6	The data does not directly support a higher cancer phenotype, especially in tissues where tumors do not normally arise. For example, despite LOH potentially occurring in different brain regions, most pediatric brain tumors are restricted to the optic pathway and brainstem, particularly in children younger than 10 years of age.	Thank you for raising this highly pertinent point that we should have included in our discussion. We have amended the manuscript accordingly. Changes to manuscript: 1) Extended discussion considering the spatial constraints of NF1-related CNS tumours vs our more widely disseminated observations (lines 188-191)
	Their comment about confirmation in experimental mouse models is puzzling. Some of the models employ biallelic Nf1 loss only, resulting in brain/nerve sheath tumors or bone	We apologise for this omission, and now discuss the data from mouse models within the constraints of the limited word count. We have highlighted the important role of mouse models in revealing the complexity of the relationship between the set of

abnormalities in select tissues despite more widespread Cre expression, while others that couple cell type-specific Nf1 loss with germline Nf1 heterozygosity reveal tumors in very specific brain regions (optic nerve, rather than cerebellum, cerebral hemispheres). These results coupled with other reports revealing differences in proliferation and differentiation that depend on the specific cell type (astrocytes and neural stem cells from different brain regions) suggest that other variables beyond bi-allelic loss underlie tumorigenesis. These observations were not considered.	mutations in a cell and its ability to form a tumour, and note the similarities with our results. Changes to manuscript: 1) Extended discussion framing our results in the context of the existing body of literature derived from mouse model experiments (lines 211-214)
Minor: NF1 is not a disorder that predisposes to neuroectodermal tumors, as many of the tumors are not of neuroectodermal origin. This factual issue should be corrected.	The cancers that are most enriched in patients with neurofibromatosis type 1 are (ordered by odds ratio in Table 2 of Landry et al [1]:  - MPNSTs (OR 9043) - Low grade glioma (OR 5473) - Osteosarcoma (OR 407) - ERMS (OR 320) - GISTs (OR 272) - Pheochromocytoma (OR 126) - High grade glioma (OR 82) - Meningioma (OR 56.7) Of these eight, six are commonly considered to be derived from neuro-ectoderm [2]:  - MPNST (Schwann cells > neural crest) - Low grade glioma - GIST (interstitial cells of Cajal > neural crest) - Pheochromocytoma (neural crest) - High grade glioma - Meningioma (from arachnoid, which is neural crest) We recognise that cancers that are not derived from neuroectoderm are also enriched in patients with neurofibromatosis type 1, albeit less strongly. We have therefore amended the manuscript in line with the reviewer's suggestion.

		Changes to manuscript: 1) Softened our description of the link between neurofibromatosis type 1 and neuroectoderm-derived tumours (lines 65-67) 1. Landry JP, Schertz KL, Chiang YJ, et al. Comparison of Cancer Prevalence in Patients With Neurofibromatosis Type 1 at an Academic Cancer Center vs in the General Population From 1985 to 2020. JAMA Netw Open. 2021;4(3):e210945. Published 2021 Mar 1. doi:10.1001/jamanetworkopen.2021.0945 2. Schoenwolf, G. C., Bleyl, S. B., Brauer, P. R. & Francis-West, P. H. Larsen's Human Embryology. (Elsevier, 2021).
	It is not clear what the “spindle cell lesion” represented. More information should be provided.	We apologise for the lack of detail. In this revision we show the histology of this lesion in the supplement, with a description in the figure legend. This lesion was originally sampled as a probable neurofibroma. Its appearances microscopically fell short of those necessary for such a diagnosis though and a broader designation of “spindle cell lesion” was deemed more appropriate by the supervising Consultant Pathologist (T.J.). Changes to manuscript: 1) Supplementary Figure 1
	Why do the authors believe that the spleen is the origin for JCML, rather than the bone marrow?	The Peer Reviewer is of course right in stating the JMML is a bone marrow leukaemia, and we apologise for our clumsy wording. What we tried to allude to is the long standing question over splenomegaly which in JMML, unlike in other leukaemias, is an obligatory diagnostic feature. We adjusted the manuscript accordingly. Changes to manuscript: 1) Clarified the relationship between the spleen and JMML (lines 171-172)

Referee 3/4 – Germline cancer susceptibility, tumorigenesis

#	Comment	Reply
3.1	Studies profiling various normal tissues now routinely identify somatic mutations commonly observed in cancer. While these are attributable to environmental exposures or endogenous mutational processes, currently, it is unclear if these mutational patterns are also observed among patients harboring inherited pathogenic variants in DNA repair genes. Here, Oliver et al., have conducted a truly impressive study profiling a vast number (~900) of normal tissue specimens from a pediatric patient with germline NF1 mutation as well as from two patients with sporadic brain tumors. They make several interesting observations that are unique to the germline patient. First, they report an acquisition of 2nd-hit (NF1 somatic mutation) among several histologically normal tissues of the germline NF1 mutated patient, a pattern that is not observed among sporadic tumors. Second, they perform histological and radiological evaluations and conclude that the NF1-null tissues are non-transformed. Third, they observe that some mutations were shared across tissues and likely arose during early development. Overall, this is a remarkable finding, albeit consistent with our intuition that predisposed cells are waiting for the one somatic hit for clonal expansion and pre-neoplastic transformation. The sheer number of distinct tissues in which these NF1-biallelic clones were observed in this patient has important implications in understanding the NF1 syndrome mediated etiology and management. This study also has important implications towards spurring interest in looking at other germline predisposition genes	Thank you for these kind comments.

	(eg: BRCA1/2 or MMR genes). Interestingly, a study profiling normal tissues in Lynch Syndrome patients did not identify either a second hit in the MMR gene or the MMR phenotype (Lee et al., PMID: 35581206). It is possible that there are gene specific differences in presentation of this phenomenon.	
3.2	Major Issues: 1. The NF1 mutations in the normal tissues are foundational to the discovery in this paper. It is critically important to present robust evidence supporting these mutations, especially given that the overall sequencing coverage is quite low (~28x). The VAF of some mutations reported here (in Figure 2E) is <10% suggesting that some NF1 mutations may be supported by 1-3 reads. The authors need to demonstrate clearly (in main or supplementary figures) that these are not artifacts or mutations observed in high mutational burden tumors (given that NF1 is a very long gene).	Thank you for raising this concern. To clarify, all NF1 point mutations found in the child with neurofibromatosis type 1 were found in one sample with at least four (subs) or five (indel) supporting mutant reads. In the old Figure 2E, the other samples shown were only determined to share the mutation after genotyping. This has been clarified with the addition of supplementary table 16. When determining if the small number of supporting reads in other samples represented a genuine mutation or artefact, we tested to see if the frequency of mutation at the locus exceeded the locus specific error rate, using a statistical model. This is an approach Sanger has developed several years ago (originally implemented in the SHEARWATER algorithm (Martincorena I, Roshan A, Gerstung M, et al. (2015). High burden and pervasive positive selection of somatic mutations in normal human skin. Science (2015)), which enables us to call mutations that present in only few reads in areas of the genome where the artefact rate is low. Put simply, if the calls were artefactual, we would see mutant reads at those positions in tissues from the other two children as well. We have clarified this in the Methods section. We would also like to highlight that we discovered second NF1 mutations through three independent sequencing approaches: bulk tissue WGS; bulk tissue exome sequencing; and WGS of microdissections (which is generated via a separate library construction protocol). To further validate our finding, we have for this revision interrogated the tissues of the original three donors with Duplex sequencing of the NF1 gene which enables us to call mutations at the resolution of single DNA molecules. The data show that non-synonymous mutations in NF1, especially truncating variants, are enriched in the child with neurofibromatosis type 1 (Figure 2f), with clear statistical evidence of

		positive selection for non-synonymous mutations. Finally, in this revision we were able to extend our findings into normal tissues (peripheral nerve, muscle, blood) of nine individuals with neurofibromatosis type 1 who had undergone sarcoma resections. We subjected these tissues to ultra high depth duplex sequencing of NF1. These data again showed great enrichment of non-synonymous NF1 variants in these individuals (Figure 2g), with a preponderance of mutations seen in nerves (where the cellular material is derived from neural crest derived Schwann cells) compared to muscle and blood. Changes to manuscript: 1) Supplementary table 16 2) Figure 2f 3) Figure 2g 4) Re-written variant calling, variant filtering and genotyping steps within Methods
3.3	2. The authors describe that they have calculated a “locus-specific error rate” and called mutations that were confidently above this threshold. However, the methodology is unclear (lines 299-303).	We apologise for the lack of clarity. We addressed this point together with the point above.
3.4	Similarly, it is particularly difficult to infer what ‘early variant genetic similarity index’ means. Moreover, it is unclear how the ‘locus-specific error rate’ is calculated (line 299 of methods).	Again, we apologise for the lack of clarity. The data shown in the previous Figure 2f were meant to ask the question ‘Are all NF1 mutant cells derived from a common embryonic lineage?’ To answer this, we compared the number of shared mutations between all pairs of samples, and looked at whether sample pairs that are null for NF1 share more mutations than samples that have retained a wildtype copy. The ‘early variant genetic similarity index’ was not an easily interpretable metric, and in the updated figure (Figure 2e), we now show the number of shared variants. While in our initial manuscript, we found that NF1 null tissues were no more closely related to one another than NF1 wildtype tissues, in this revised manuscript we have compared tissues that share specific second hits. In this improved analysis we can see that some tissues with the same mutation are more closely related than

		those without. We have updated the figure and our statements in the main text accordingly. Changes to manuscript: 1) New Figure 2e 2) Removal of old Figure 2f 3) Re-writing of statements relating to sharing of mutations (lines 138-150)
3.5	3. The authors seem to use “transformed” and “neoplastic” synonymously and imply that non-neoplastic normal tissues must be non-transformed. However, the high VAFs for somatic NF1 mutations in many non-neoplastic normal tissues suggest strong positive selection. (It is hard to argue that a ~30% VAF NF1 clone (cerebellum) is not “transformed”.) Presumably all mutant clones that are detectable from bulk WES or WGS have undergone prior positive selection. Further, by considering neoplastic transformation as binary (i.e., “normal” and “neoplastic” in Figures 1A-B), this fails to consider intermediate stages where the “non-transformation of NF1” may be either because the clonal expansions are in very early stages of neoplastic transformation and/or are not detectable via histology or radiographically. Perhaps the authors could consider adopting language supporting the notion that these NF1 null clones may be on path of pre-neoplastic transformation.	This is a valid concern which alludes to two issues: a lack of semantic clarity and a more nuanced discussion of what NF1 null tissues represent. We have amended the manuscript accordingly. Changes to manuscript: 1) We now state (supported by dN/dS analysis) that NF1 mutations are under positive selection in the normal tissues of patients with neurofibromatosis (lines 42, 168, 179-180) 2) We use neoplasm to mean a clonal expansion with a phenotypic change, including malignancies but also benign lesions such as cafe au lait spots, whereas transformation we reserve to malignancy. For clarity, we have replaced ‘transformation’ with ‘transformation to a frank malignancy’ in the discussion (line 206) 3) We have added an extended discussion where we consider the fate of these NF1 null clones and speculate that other factors affect their risk of transformation (lines 209-214)
3.6	4. The disease presentation, genomics, histopathology and treatment histories for each of the three patients is not described in the paper (line 53). Based on Table S1, one patient had tumors in six different regions of the brain. But that is not described in text or depicted in any of the figures. And what are the somatic NF1 mutations and other	We apologise for the lack of clarity. The gliomas themselves were not a focus of this work, beyond us developing a method to exclude tumour contamination as the cause of our observations. Nonetheless, we should have provided more information. We briefly summarise the treatment of the three patients with the main manuscript. We have added a new Supplementary table 2 that describes the number and distribution of tumour biopsies examined for driver mutations. This will

	co-occurring driver alterations in these diseases (eg: TERT, ATRX, CDKN2A, TP53, PTEN, PIK3CA)?	hopefully provide a clearer sense of the breadth and depth of our tumour sampling. Furthermore, the new Table 1 lists all the genes subject to driver events in the gliomas, compared to other neoplasms and normal tissues. Supplementary tables 11-15 provides a complete list of all the driver mutations found within all of the gliomas in the three children. Changes to manuscript: 1) New Supplementary Table 2 added 2) Table 1 added
3.7	And how are the nearly ~450000 mutations reported in the supplementary table evaluated for driver status?	Again, we apologise for the lack of detail. To evaluate driver status, we adhered to convention, as implemented at Sanger for over a decade and formalised in many publications, which we detail in the revision of this manuscript. Changes to manuscript: 1) We have provided a detailed description of our driver mutation discovery and annotation in the Methods.
3.8	5. The germline NF1 mutation in the patient with neurofibromatosis does not appear to be specified anywhere in the main text or figures.	We apologise for this omission. This child has an essential splice site (c.3113+1G>A) germline mutation in NF1. We have now included this prominently in the main text. We have also provided the germline NF1 mutation for the nine additional adult patients we included in the revised manuscript (Supplementary Table 17). Changes to manuscript: 1) Supplementary Table 17 2) Inclusion of the germline NF1 mutation in the main text (line 76).
3.9	6. The table in Figure 1E lists the numbers of specimens collected but it isn't clear how many were from the primary tumors and which were from normal tissues. For readers not familiar with the intricate brain anatomy, it will be helpful to have a schematic showing the regions of the brain that were profiled for each patient and the numbers	Thank you for this suggestion. For simplicity's sake, we have provided another version of Figure 1e where we include all biopsies that passed QC and were deemed to have high enough glioma purity to examine them for driver mutations (Supplementary table 2, Methods). We have also provided a new Table 1 where the reader can see none of these tumour biopsies sustained any further NF1 point

	of LCM/bulk specimens collected in each region. Moreover, this spatial depiction is important to: (1) understand potential tumor infiltration into normal tissues profiled here, and (2) to understand the NF1 mutations that arose in normal tissues. As this study is poised to garner interest across the fields (genomics, bioinformatics and clinical oncology), it is important to make the anatomical descriptions of the disease as well as the specific regions sampled as interpretable as possible.	mutation of any kind in PD51122, beyond the initial LOH event, whereas the normal tissues endure multiple loss of function variants. We are extremely confident the tumour infiltration into normal tissue does not account for our observations here for a number of reasons:  1) The tissues are histologically normal (Supplementary figure 3) 2) Our method for detecting tumour infiltration is highly sensitive and was able to detect it down to <1% (see column S in Supplementary table 3). This approach was supported by simulated data (Methods). Many of these NF1 second hits are found in tissue with 0% tumour involvement and are sufficiently large that even an undetectably small infiltration of tumour could not account for them, e.g. 56% of cells are NF1 null in PD51122b_lo0018. 3) All the glioma cells in the child with neurofibromatosis type 1 had no functioning copies of NF1 left and therefore there was no selective advantage in further mutating it. This is confirmed by the absence of any NF1 point mutations in it in Table 1. 4) The previous Figure 2a (the new Figure 2c) shows that somatic mutations in NF1 were present in anatomically disparate parts of the central nervous system of patient PD51122. 5) In this revised manuscript, we have included an extension cohort of adults with neurofibromatosis type 1 who had had normal tissue removed with sarcoma resections. Duplex sequencing of these normal tissues revealed >2 inactivating NF1 mutations per individual in several individuals. In these cases too, therefore, it is highly unlikely that NF1 mutations are the result of contamination from the tumour. Changes to manuscript:  1) Supplementary table 2 2) Table 1
3.10	7. The premise for doing bulk tissue sequencing “to extend and validate findings” needs more clarity (line 75 of main	We have substantially rewritten our experimental outline to incorporate the additional duplex sequencing data from the original three children and the

	text and line 13 of supplementary). How are they selected? Are they predominantly normal tissues in regions where NF1 mutations were detected? Moreover, line 76 suggests bulk tissue WES for 180 specimens were performed but Figure 1E indicates that the WES samples are LCMs	additional cohort of individuals with neurofibromatosis. We hope this, as well as the Methods section, provide a clearer explanation of our rationale for the approach taken. In short, all bulk biopsies acquired during the autopsy were subject to bulk WGS with a few exceptions. Those that were not included normal tissues where we were confident, on the basis of prior literature, that clonal or oligoclonal cell populations could be isolated by microdissections, such as the colon. In such samples, deep sequencing of mixed, polyclonal bulk colonic tissue would be of limited additional benefit. The other major group of bulk tissues where bulk WGS was not pursued were those taken at the macroscopic interface of tumour and normal tissue. Here, we felt that microdissection would enable us to capture both tissues independently, rather than the low tumour purity readout we would acquire from bulk sequencing. The central nervous system tissues, where we found all the deleterious NF1 variants by the standard sequencing and variant calling approach, was a focus of our sampling at post mortem and then when subsequently sequencing the acquired tissues, even before our discovery of the NF1 second hits. The narrative of pervasive second NF1 hits within the central nervous system emerged within the WGS data from microdissected tissues but the bulk WGS and LCM WES substantiated and further validated our observation of this phenomenon. It should now be clear that WES was only performed on microdissected tissues (line 88). We apologise that the prior wording was misleading. Changes to manuscript:  1) Further clarification in the Methods 2) The description of the experimental outline in the main text has been updated (lines 85-93).
3.11	Overall the methods are difficult to read and understand. It would be helpful to present each section with first describing the premise of the methods.	We apologise for the lack of clarity. We have rewritten the methods and streamlined some bespoke analytical steps to facilitate reproducibility of our work.

		Changes to manuscript:  1) Substantial re-writing of the Methods section.
3.12	9. Line 79 refers to low-level tumor contamination in normal tissues. The methods and interpretation are unclear. Was the original tumor sequenced deeply enough to capture the full spectrum of tumor specific truncal “driver” mutations, especially the NF1 somatic mutation? The authors should explain why that is not sufficient to identify and eliminate tumor infiltrating tissues. Moreover, did the authors evaluate contamination in only the specimens proximal to the tumor or all tissues, even those that are distal to the tumor.	A total of 70 regions, bulk and microdissected, underwent whole genome sequencing and were used to identify driver mutations in the glioma from the child with neurofibromatosis type 1. The combined sequencing depth of these biopsies was extraordinarily deep (~2400X), making it impossible for us to miss truncal driver events on the basis of insufficient sequencing depth. Where tumour infiltration could have proven an issue was in the presence of subclonal NF1 mutations as these may be absent in the purer areas of tumour sampled. However, as we discussed in 3.9, this cannot explain our findings. We considered all truncal glioma substitutions, drivers and passengers, when looking for evidence of tumour infiltration as longer catalogues of mutations would improve our sensitivity for low level tumour detection, as was supported by our simulated data (Methods). All of the loci used in this analysis for each glioma were examined in every sample from the patient, irrespective of how far it was away from it. Changes to manuscript:  1) Clarification of the exclusion of tumour mutations in the Methods
3.13	10. Figure 2C. The authors interpret that there was no elevated mutational burden in NF1-null tissues. However, at least for a few of the NF1-null specimens, the overall substitution count was high (and highest among the bulk WGS specimen).	Based on the Reviewer’s comments, we have re-interrogated the data. The median substitution burden of NF1-heterozygous normal CNS samples from PD51122 analysed by WGS of microbiopsies was 8 substitutions, while that of NF1-null heterozygous normal CNS samples from the same child was 15 mutations. This is a statistically significant difference using either a Wilcoxon rank sum test or a linear mixed effects model that accounts for potential confounders (coverage and experimental approach (microbiopsy vs bulk) as fixed effects, and the piece of tissue from which a sample derives as a random effect). The effect size associated with

		NF1-null status (which is also of 7 mutations in the linear model), however, is not particularly biologically meaningful with regards to cause or effects of the NF1 second hit. Such a small increase in mutation burden could not be a cause of the NF1 second hits in these tissues, nor is it compatible with a recent clonal expansion. With a recent clonal expansion, we would expect all the mutations that occurred in the cell prior to its clonal expansion to be detectable. Given the postnatal somatic mutation rate of most tissues is 10-50 mutations per year (including a rate in glia of 27 substitutions per year (Ganz et al. Contrasting patterns of somatic mutations in neurons and glia reveal differential predisposition to disease in the aging human brain, bioRxiv, doi: 10.1101/2023.01.14.523958) and a rate in neurones of 17 substitutions per year (Abascal et al., Somatic mutation landscapes at single-molecule resolution, Nature, 593,405-410 (2021)), and the prenatal rate is usually higher, a recent clonal expansions should result in a mutation burden on the order of 100 mutations even in a child; this is a far cry from the excess of 7 extra mutations that we observe. We have altered our statement in the manuscript to limit ourselves to excluding a recent clonal expansion. Changes to manuscript:  1) Lines 118-119 2) Description of the linear model used in the Methods section.
	11. Figure 2E and Table S2: Mutations identified in bulk WGS should be highlighted distinctly from those that were LCM derived. For mutations observed in multiple tissues, it is likely that the proximity of the biopsied regions (eg: pons/medulla or parietal cortex/cerebellum) could contribute to a mutation observed in LCM in one tissue and bulk of the other. In fact, this appears to be the case for R304 and I679*. Both these were identified in LCM of one tissue and bulk WGS of a proximal tissue. The authors should re-evaluate these mutations and further exclude specimens that might have tissue infiltrates from	In this revision we present these data in a dedicated Table (Table 1) to provide more detail on the source of each mutation. Our interpretation of second NF1 mutations that are seen in multiple locations is precisely what the Reviewers suggest, namely that the mutation represents a clone that spans different anatomical areas, in some cases neighbouring areas, in others, distant regions of the brain. For example, the R304* variant is seen in two independent chunks of the left parietal cortex. Although not immediately adjacent, the two biopsies broadly represent the same anatomical territory. However, the same mutation is also seen in the cerebellum which is anatomically distant from the parietal cortex, and separated from it by the tentorium. This mutation therefore

	neighboring regions.	may represent a second NF1 hit in an early neuroectodermal lineage or, as an independent R304* mutation that evolved in parallel. In an attempt to answer this question in the revised manuscript, we have added a new analysis that compares the number of shared mutations between all pairs of samples, and looked at whether sample pairs that share the same second hit in NF1 share more mutations than other sample pairs. For p.R304* and p.I679fs*21, samples shared more mutations than other pairwise comparisons within the CNS (Figure 2e), suggesting that these may represent the same early event. In contrast, brain and spleen samples that share the Y489C variant were not closely related to one another, indicating that this variant is more likely to have occurred twice independently. Changes to manuscript: 1) Addition of Figure 2e 2) Revised interpretation of the data in light of this analysis and the insights gained from the additional cases sequenced (lines 138-150)
3.14	12. Line 124: If Y489C occurred pre-gastrulation, wouldn't it have been more clonal in the spleen (Figure 2E) and also seen in other tissues with related origin? As this is a recurrent hotspot, could this mutation have arisen independently in these two tissues, especially if it was not observed in other brain and extra-cranial tissues?	We entirely agree, and we discuss this possibility in conjunction with quantification of sharedness in this revised manuscript - please see above. A similar point is raised and addressed in 1.3. Changes to manuscript. 1) lines 138-150
3.15	Minor issues: 1. In line 35, consider changing "which may explain the tissue specificity" to "which may be explained by the tissue specificity"	We have altered this sentence of the abstract to reflect the Reviewers' comment. Changes to manuscript 1) lines 44-45

3.16	2. Figure 2D and S5. A legend is needed to describe that the red histogram corresponds to the haplotype with the NF1-mutant allele.	We apologise for this oversight. This is now done (old Figure 2D is now Supplementary figure 6). Changes to manuscript:  1) Alterations to legends of supplementary figures 5 and 6 (new version).
3.17	3. Line 83: no description or interpretation of the mutation burden or substitution signatures were provided.	Owing to constraints in word counts, we omitted this from the original manuscript and only discussed the mutation burden in terms of timing a clonal expansion (line 119, Figure 2b (new version)). However, all mutations are provided in the Supplementary tables, should the reader wish to examine the mutational profiles of our cases. Overall the landscape of signatures was “as expected” - of some interest perhaps was that we found UV light signatures in the skin of children and signatures associated with Helicobacter in stomach. Changes to manuscript.  1) Lines 118-119. 2) Methods section describing testing for a recent clonal expansion.
3.18	4. Line 113: loss of the germline NF1 may be by chance and is most likely unremarkable and definitely does not constitute a ‘reversion’.	We have amended the manuscript according to this concern. Changes to manuscript:  1) Changed description of the observed loss of chromosome 17 bearing the germline mutated copy of NF1 within normal tissue (lines 132-135)

Rebuttal for manuscript 'Cancer-independent, second somatic *NF1* mutation of normal tissues in neurofibromatosis type 1'

We thank the Editor and Reviewers for their comments, to which we have responded below.

Reviewer #1

REVIEWER COMMENT	AUTHORS' RESPONSE
The authors have addressed all my comments. I would like to congratulate them with a wonderful paper. I have only two small points for their consideration:	We thank the reviewer for their kind words.
- Is it possible to phase some of the 29 non-synonymous mutations found by duplex sequencing to the allele which is not affected by the germline NF1 mutations? As pointed out in the manuscript, the authors know which SNPs are located on the germline mutated allele thanks to the LOH event. Of course, not all the 29 non-synonymous mutations can probably be phased, but for the ones that can be, it would be good to validate that these are indeed located on the WT allele.	At six loci either the mutant read or its mate can be phased. In all six cases, the truncating NF1 mutation is out of phase with the germline mutation (i.e. on the wildtype allele). Therefore, where phaseable, we can demonstrate that both alleles of NF1 are inactivated.
- The title of the graphs in Figure 2a is on the wrong side (right instead of left side of the graph).	Thank you. This has now been corrected.

Reviewer #2

REVIEWER COMMENT	AUTHORS' RESPONSE
The brief report by Oliver and colleagues presents whole genome sequencing of two children with diffuse midline glioma and one with neurofibromatosis type 1 (NF1). They use a number of bioinformatic methods to reveal multiple independent somatic NF1 driver mutations in histologically normal tissues, which they infer are lineages prone to neoplasia. They conclude that these preliminary findings may explain the tissue specificity and variable penetrance of oncogenesis in NF1.	We thank the reviewer for re-appraising our revised manuscript.
It is unclear why the authors conclude Y489C is a mutational hotspot. There is no clear evidence for mutational hotspots in the NF1 gene, either in the context of neurofibromatosis or in sporadic cancers.	Thank you for highlighting this. In our previous response to Reviewers we presented evidence that Y489C is a driver mutation. In addition to the recurrence of this particular mutation in COSMIC, we gave supporting evidence from the Karolinska Molecular Tumour Board's in silico prediction tool, and Clinvar data. Evidence in Clinvar of the mutation's pathogenicity includes families with a clinical diagnosis of neurofibromatosis type 1 who have Y489C as the only detected NF1 germline mutation that segregates with the phenotype, as well as being found as a de novo variant. Finally, we presented transcriptomic studies that show that this variant results in altered splicing that results in a deletion of part of exon 13. Thus, this missense variant should be considered pathogenic. As the reviewer highlights, however, the term 'mutational hotspot' may be controversial. We have therefore removed the term 'hotspot' and outlined our reasoning to suggest that Y489C is oncogenic. Changes to manuscript:

	1) l.105-107: changed from ‘occurred in a mutational hotspot’ to ‘were likely oncogenic based on recurrence, in silico predictions, correlation between genotype and phenotype, and functional studies’ 2) l. 672-673: changed from ‘mutational hotspot’ to ‘residues that are mutated more frequently’
Why is it unlikely that additional mutations would occur in tumor cells? There are descriptions of additional (rare) somatic mutations in PTEN, BRAF (KIAA1549:BRAF), and FGFR1 in NF1-associated brain tumors.	We meant that further mutations in NF1 are unlikely in the tumour as both copies have been inactivated. We agree that there are additional mutations in other genes in NF1-associated brain tumours, and to avoid confusion we have specified that we are talking only about NF1. Changes to manuscript: 1) l. 116-117: changed from ‘as the second NF1 hit in the glioma was loss of heterozygosity, it is unlikely that further NF1 mutations would occur in tumour cells’ to ‘as both copies of NF1 in the glioma were already inactivated, there should be no selection pressure for further loss-of-function mutations in NF1 in tumour cells’
Is there evidence for a founder mutational effect, where regionally related cells share the same somatic mutation?	This is a pertinent point. We have attempted to assess this at two levels:  1) looking within a piece of tissue, which may indicate clonal expansions 2) looking between pieces of tissue from anatomically close areas, which may indicate shared embryological ancestry. Within a piece of tissue, we found NF1 variants in up to 56% of cells (based on allele frequency), which suggests that the NF1 variant is under positive selection and that those cells that bear it have a survival or proliferative advantage. This is also consistent with our dN/dS analysis results. Mutations shared between pieces of tissue may represent enormous postnatal clonal sweeps (which is unlikely if there are wildtype areas between mutant ones), shared embryological ancestry, or convergent evolution with the same mutation arising independently more than once. We have attempted to distinguish between the latter two scenarios in the analysis that we present in figure 2E. We show that in some cases, tissues that share the same mutation in NF1, share more variants across the genome than expected by chance, implying a shared ancestry. For example, the one NF1 variant (p.l679fs*21) is shared between the pons and medulla, anatomically adjacent structures. Our analysis suggests that a “founder” cell acquired this mutation and expanded across both regions.

	Relevant sections in manuscript: 1) l. 112-114: ‘ The variant allele frequencies (VAFs) of second NF1 hits indicated clone sizes as large as 56% of cells (clone size = 2 x VAF) in a microdissection (hundreds of cells) and 19% in a bulk tissue (a macroscopic piece of tissue).’ 2) l. 112-150: ‘There were two possible explanations: either the shared variants arose prior to seeding of the anatomical areas in which the mutations were found, or mutations appeared independently in different tissues. We could establish which scenario was more likely by comparing the total number of mutations across the genome that were shared between affected tissues: those with a common developmental root would possess more. For NF1 mutations that spanned only regions of the brain (p.R304*, p.I679fs*21), affected tissues shared significantly more mutations with each other than unaffected with other brain regions did ($P < 0.001$ for both mutations, permutation test; Fig. 2e, Methods), implicating a common ancestor in their development’
While leukemic cells do get trapped in the spleen, this may not be their cell of origin. The sentence should be revised.	We have corrected this. Changes to manuscript: 1) l. 175-177: changed from ‘This finding was of interest given that neurofibromatosis type 1 predisposes to juvenile myelomonocytic leukaemia, which always (as per the diagnostic definition) involves the spleen²²’ to ‘This finding was of interest given that neurofibromatosis type 1 predisposes to juvenile myelomonocytic leukaemia, which always (as per the diagnostic definition) involves the spleen²², whether through entrapment of leukaemic cells or as their organ of origin’.
The sentence about “transformation to a frank malignancy” should be revised to reflect the fact that the majority of tumors arising in the setting of NF1 are non-malignant (benign) tumors.	We thank the reviewer for pointing this out. Changes to manuscript: 1) line 211: change from ‘frank malignancy’ to ‘discernible tumour’.

Reviewer #3/4

REVIEWER COMMENT	AUTHORS' RESPONSE
The revised study is now substantially improved and addresses many of my original comments. The authors have generated new data and have now quantitatively established evidence for positive selection of clones harboring somatic NF1 in normal tissues of predisposed individuals. This is a very interesting finding and supports our intuition and will be a significant addition to literature. Some minor concerns mainly surrounding the clarity with which the results are presented remain.	We thank the Reviewers for their comments on all iterations of the manuscript. Specific replies are provided to their most recent comments below.
1. The level of detail with which the evidence for the mutations highlighted in the study is underwhelming. Table S6 does not have the VAFs or the read coverage information. It is helpful to include the VAFs in Figure 2C. Currently, there is no way to understand the evidence supporting the genotyping VAFs shown in Figure 2E. For example, I679fs* was originally detected at 10% in only one of the non-neoplastic tissues. But in the genotyping step, the authors report detecting this mutation at >>10% VAF in 3 tissues in which the original mutation calling pipeline did not detect this event. This may be because the locus-specific error rate is very low. But is it possible that the overall background rate of mutations is very high in those tissues in which these high VAF	The VAF and read coverage for each of the driver point mutations called in the WGS and WES data from the autopsy cases can be found in Supplementary tables 11 and 12. Nonetheless, we have now added this information to Supplementary tables 5 and 6 for completeness. The VAFs have also been added to Figure 2c. We apologise for the confusion regarding the VAFs in Figure 2d. The sensitivity and specificity of variant calling varies between different tools. Indels are a notoriously difficult group to call although Pindel compares favourably to others when benchmarked (see below, labelled "Sanger", taken from ref. 1):

mutations are genotyped?

With regards to the mutation the Reviewers specifically mentioned, Pindel failed to call the indel in all six samples, either because there were insufficient supporting reads, as determined by Pindel, or an insufficient number of those reads passed Pindel's internal flags.

Once one has determined that an indel is present, it remains a challenge to determine its VAF accurately, certainly compared to substitutions. This is due to the inherent challenges of mapping the indel-containing reads. Exonerate attempts to overcome this by re-interrogating the BAM file to identify further reads that support a variant which might not have been considered during the initial variant calling. It also disregards reads that support neither the mutant or wildtype allele. As a consequence, it is likely that the genotyped indel VAF is noisy and may differ from that estimated by

the original variant caller. For these reasons, we do not use indel VAFs / data when we need VAF precision for analyses. For example, when determining clone sizes we use substitutions.

After reflecting on the Reviewers' important point, we think it is more accurate to portray the data in Figure 2d as a binary mutation present/absent table instead. In Figure 2c, the VAFs presented are those directly estimated by Pindel and thus are consistent with the calls contained within the supplementary tables.

To further reassure the Reviewers of the validity of our mutations from the original autopsy WGS/WES experiment, we would point them to the examples of 17q LOH shown in supplementary figure 5 and the images below which are taken from Jbrowse, the genome browser. These show the sequencing reads supporting each point mutation in the original sample it was called in (red box highlights the mutated locus).

NF1 p.K233fs*48 (note that it phases correctly with the adjacent SNP)

NF1 p.R304*

NF1 p.Y489C

NF1 p.L679fs*21 (note that it phases correctly with the adjacent SNP)

Changes to manuscript

- 1) VAFs and read coverage added to Supplementary tables 5 and 6
- 2) Figure 2c amended to include VAFs
- 3) Figure 2d changed to a table

1. The ICGC/TCGA Pan-Cancer Analysis of Whole Genomes Consortium. Pan-cancer analysis of whole genomes. *Nature* 578, 82–93 (2020). <https://doi.org/10.1038/s41586-020-1969-6>

2. Methods have been substantially revised but there is a still lot missing. The authors need to describe how they performed the permutation test for Figure 2E (for example, how is a

We apologise for not explaining the permutation test used in Figure 2e more clearly. This has now been added to the Methods section.

permutation test performed with just one data point for NF1 p.Y489C), or the dN/dS calculations in Figure 2F, G. For indels, the authors report using “Exonerate”. But how was genotyping performed for SNVs?	In short, for each NF1 mutation, we divide samples into two groups: those with the mutation (test tissues), and those without the mutation (control tissues). We count the number of shared mutations between each pair of test tissues, and take the average across all pairs. Let the number of test tissues be called n. We then sample n control tissues without replacement 1,000 times. For each sample of control tissues, we calculate the average number of shared mutations between all possible pairs. This generates a distribution of sharedness that we can compare to the test group. The P value was determined by the number of draws where the control data mean was greater than that observed in the test data (one-sided test). If the test tissues share more mutations than the vast majority of the control group samples, it is likely that the samples in the test group share a more recent common ancestor, and so the NF1 mutation may have occurred in development. Each pair of samples that share an NF1 mutation results in a single comparison and so a single data point. The question that the test asks, in the case you reference, is what proportion of 1,000 random pairwise comparisons from the control group show a greater genetic sharedness (one-sided test) than that seen between the two samples with p.Y489C? The explanation of the dNdS analysis has been expanded within the Methods section. We have also added a note on the independence of mutations. Genotyping for SNVs was performed using pileup within vafCorrect. This is now described within the methods. Changes to manuscript  1) Description of permutation test within the Methods (l. 760-783) 2) Expansion of dNdS analysis explanation in the Methods (l. 551-554) 3) Description of SNV genotyping within the Methods (l. 433-435)
3. Regarding 2E, why are there a median of 2 shared mutations between all pairs of CNS vs. CNS	Mutations are acquired in our cells from the first cell division. These earliest mutations inevitably pervade large amounts of our tissue. The tissues included in this comparison are broadly oligo- and

and CNS vs. WES samples? Isn't the expectation that there are 0 shared mutations between clonally unrelated tissues? Do the authors hypothesize that these mutations are independently arising or could they be some type of artifact?

polyclonal, drawing from many cell lineages that would have possessed different early (mosaic) mutations. The median of ~2 shared substitutions between two random samples likely reflects the shared pre-gastrulation heritage of subsets of the cells in each sample.

Of course, one cannot entirely exclude the possibility that rare artefacts passed our stringent filtering but it would have no impact on our interpretation: artefacts are just as likely to affect a comparison between the random tissues as they would those samples with a shared second *NF1* mutation.